# Exploring Adversarial Robustness of Deep State Space Models

**Biqing Qi[1,2], Yiang Luo[3], Junqi Gao[4],\*, Pengfei Li[4], Kai Tian[1], Zhiyuan Ma[1], Bowen Zhou[1,2],\***

[1] Department of Electronic Engineering, Tsinghua University,
[2] Shanghai Artificial Intelligence Laboratory,
[3] Department of Control Science and Engineering, Harbin Institute of Technology,
[4] School of Mathematics, Harbin Institute of Technology
{qibiqing7,normanluo668,gjunqi97,lipengfei0208}@gmail.com,
tiankaidyx@163.com, {mzyth,zhoubowen}@tsinghua.edu.cn

## Abstract

Deep State Space Models (SSMs) have proven effective in numerous task scenarios but face significant security challenges due to Adversarial Perturbations (APs) in real-world deployments. Adversarial Training (AT) is a mainstream approach to enhancing Adversarial Robustness (AR) and has been validated on various traditional DNN architectures. However, its effectiveness in improving the AR of SSMs remains unclear. While many enhancements in SSM components, such as integrating Attention mechanisms and expanding to data-dependent SSM parameterizations, have brought significant gains in Standard Training (ST) settings, their potential benefits in AT remain unexplored. To investigate this, we evaluate existing structural variants of SSMs with AT to assess their AR performance. We observe that pure SSM structures struggle to benefit from AT, whereas incorporating Attention yields a markedly better trade-off between robustness and generalization for SSMs in AT compared to other components. Nonetheless, the integration of Attention also leads to Robust Overfitting (RO) issues. To understand these phenomena, we empirically and theoretically analyze the output error of SSMs under AP. We find that fixed-parameterized SSMs have output error bounds strictly related to their parameters, limiting their AT benefits, while input-dependent SSMs may face the problem of error explosion. Furthermore, we show that the Attention component effectively scales the output error of SSMs during training, enabling them to benefit more from AT, but at the cost of introducing RO due to its high model complexity. Inspired by this, we propose a simple and effective Adaptive Scaling (AdS) mechanism that brings AT performance close to Attention-integrated SSMs without introducing the issue of RO. Our code is available at Robustness-of-SSM.

## 1 Introduction

Deep State Space Models (SSMs) represent a promising emerging model architecture inspired by traditional linear state space models [1]. These models capture temporal dependencies in sequences via structured state space transitions [2, 3, 4]. By leveraging their inherent recurrent nature and linear discretized transmission form, SSMs excel in long-sequence modeling with linear computational complexity [3, 5]. This makes SSMs highly competitive in various downstream tasks, with the potential to surpass classical architectures like convolutional networks and Transformers [6, 7, 8]. While the superior performance of SSMs on clean data has been well established across various tasks, these models may face significant security risks from meticulously crafted Adversarial Perturbations (APs) [9, 10, 11, 12, 13]in real-world scenarios. Therefore, it is essential to thoroughly explore and discuss the Adversarial Robustness (AR) of SSMs. Previous work [14] has conducted

---

\*Corresponding authors.

38th Conference on Neural Information Processing Systems (NeurIPS 2024).

preliminary robustness evaluations on visual SSMs. However, their assessment was limited to the VMamba structure. More importantly, they did not specifically evaluate and analyze the performance of SSMs under Adversarial Training (AT) [10, 15, 16]. This aspect is crucial for enhancing the AR of SSMs, as AT is the mainstream approach for improving a model's AR [16, 17, 18]. To bridge the current research gap, we aim to answer the following questions:

1. *Do many component designs that have proven effective in boosting SSM performance under Standard Training (ST) [19, 20] similarly enhance SSMs when subjected to AT?* Adversarial attacks aim to induce erroneous predictions by adding meticulously crafted small perturbations [9, 10], which may cause a more pronounced negative impact on SSM models. This is because SSMs heavily rely on sequential dependencies in inputs, causing errors to accumulate through state propagation. Consequently, it remains uncertain whether commonly employed AT frameworks, such as PGD-AT [16] and TRADES [21], can effectively improve the AR of SSMs.

2. *How do various components influence the robustness-generalization trade-off in AT?* Due to the trade-off between robustness and generalization [22, 23], AT often leads to a decrease in the accuracy of deep learning models on clean data, thus hindering their generalization. Additionally, deep learning models are prone to Robust Overfitting (RO) [24, 25], where models trained adversarially overfit to the AT samples but fail to generalize to adversarial test samples, significantly limiting the effectiveness of AT. We aim to investigate how existing structural variants of SSMs, such as diagonalized [26] and Multi-Input Multi-Output (MIMO) SSM [27], affect the robustness-generalization trade-off and RO issues in SSMs, thereby providing insights for designing more robust SSM structures.

3. *Can the analysis of various SSM components' performance in AT provide valuable insights for the design of more robust SSM structures?* Our objective is to analyze how components of different SSM variants influence model behavior during AT and to identify those that contribute positively. This analysis aims to inform adjustments that enhance the robustness and performance of SSMs under AT.

To response Q1 and Q2, we employ AT using common AT frameworks on various structural variants of SSMs. These include implementations such as Normal Plus Low-Rank (NPLR) decomposition (S4) [3], diagonalization (DSS) [28, 26], MIMO extension (S5) [27], integration of attention mechanisms (Mega) [19], and data-dependent SSM extension (Mamba) [20]. Subsequently, we conduct comprehensive robustness evaluations to assess their performance. Specifically, we obtain the following observations: 1)**A Clear Robustness-Generalization Trade-off in SSM Structures**. We observe a significant decrease in the Clean Accuracy (CA) of SSMs following AT. Specifically, for the S4 model on the CIFAR-10 dataset, the CA dropped by nearly 15% compared to ST. This indicates a pronounced robustness-generalization trade-off in SSM structures. 2) **Pure SSM structure struggles in benefit from AT. Despite expanded to data-dependent SSM, Mamba fails to show distinct superiority over the fixed-parameterized counterparts (S4 and DSS) under AT, even though all are SISO systems**. 3) **Only the incorporation of attention mechanisms significantly enhances both the CA and RA of SSMs after AT. However, this improvement also introduces a significant issue of RO.**

To response Q3, we conduct both empirical and theoretical analyses of the output error of SSMs under AP. Our findings demonstrate that fixed-parameterized SSMs have bounded output errors strictly related to their parameters, which limits their ability to reduce output errors during AT. In contrast, data-dependent SSMs may encounter an explosion of output errors. Furthermore, incorporating attention mechanisms provides additional adaptive scaling of the SSM's output error, facilitating better alignment of outputs corresponding to adversarial and clean inputs. However, this also increases complexity, raising the risk of RO. Inspired by this, we explore whether a low-complexity adaptive scaling operation could assist SSMs in output alignment while avoiding RO issues. To this end, we designed a simple Adaptive Scaling (AdS) mechanism to aid in output alignment. Our results indicate that this design effectively enhances the AT performance of SSMs and reduces output errors without introducing RO issues. We hope that our responses to the above three questions will inspire further research and development of robust SSM structures.

## 2 Preliminaries of SSMs

Given the input signal $\boldsymbol{u}(\cdot) \in \mathbb{R}^+ \to \mathbb{R}^d$ and time $t \in \mathbb{R}^+$, the general linear SSM, parametrized by an state matrix $\boldsymbol{A} \in \mathbb{R}^{h \times h}$, an input matrix $\boldsymbol{B} \in \mathbb{R}^{h \times d}$, an output matrix $\boldsymbol{C} \in \mathbb{R}^{d \times h}$ and a direct transmission matrix $\boldsymbol{D} \in \mathbb{R}^{d \times d}$ can be formalized as follows:

$$\dot{\boldsymbol{x}}(t) = \boldsymbol{A}\boldsymbol{x}(t) + \boldsymbol{B}\boldsymbol{u}(t), \tag{1}$$

$$\boldsymbol{y}(t) = \boldsymbol{C}\boldsymbol{x}(t) + \boldsymbol{D}\boldsymbol{u}(t), \tag{2}$$

where $\boldsymbol{x}(t) \in \mathbb{R}^h$ and $\boldsymbol{y}(t) \in \mathbb{R}^o$ represents respectively represent the hidden state and the output signal at time $t$.

**S4**  For the discretized input sequence $\boldsymbol{u} \in \mathbb{R}^{L \times d}$, S4 [3] broadcasts the same SSM parameters to each input dimension for state propagation, instantiating a SISO system. Specifically, they employ a bilinear method [29] for discretization to obtain discretized approximations of the parameters:

$$\boldsymbol{x}_k = \overline{\boldsymbol{A}}\boldsymbol{x}_{k-1} + \overline{\boldsymbol{B}}\boldsymbol{u}_k^{(i)}, \quad \overline{\boldsymbol{A}} = (\boldsymbol{I} - \Delta t/2 \cdot \boldsymbol{A})^{-1}(\boldsymbol{I} + \Delta t/2 \cdot \boldsymbol{A}) \in \mathbb{R}^h \tag{3}$$

$$\boldsymbol{y}_k^{(i)} = \overline{\boldsymbol{C}}\boldsymbol{x}_k, \quad \overline{\boldsymbol{B}} = (\boldsymbol{I} - \Delta t/2 \cdot \boldsymbol{A})^{-1}\Delta \boldsymbol{B} \in \mathbb{R}^{h \times 1}, \quad \overline{\boldsymbol{C}} = \boldsymbol{C} \in \mathbb{R}^{1 \times h}, \tag{4}$$

where $\boldsymbol{u}_k^{(i)}$ and $\boldsymbol{y}_k^{(i)}$ represents the $i$-th element in the $k$-th token of $\boldsymbol{u}$ and $\boldsymbol{y}$ respectively. $\Delta t \in \mathbb{R}^+$ is a fixed step size that remains constant for each time step. S4 parameterizes the matrix $\boldsymbol{A}$ using NPLR decomposition for efficient optimization. By setting $\boldsymbol{x}_0 = 0$, the output can take the following form:

$$\boldsymbol{y}_k^{(i)} = \overline{\boldsymbol{C}}\overline{\boldsymbol{A}}^{k-1}\overline{\boldsymbol{B}}\boldsymbol{u}_1^{(i)} + \cdots + \overline{\boldsymbol{C}}\overline{\boldsymbol{A}}\overline{\boldsymbol{B}}\boldsymbol{u}_{k-1}^{(i)} + \overline{\boldsymbol{C}}\overline{\boldsymbol{B}}\boldsymbol{u}_k^{(i)}. \tag{5}$$

This can be achieved by a parallelized efficient convolution computation: $\boldsymbol{y}^{(i)} = \overline{\boldsymbol{K}} * \boldsymbol{u}^{(i)}$, where the convolution kernel $\overline{\boldsymbol{K}} = \left(\overline{\boldsymbol{C}\boldsymbol{B}}, \overline{\boldsymbol{C}\boldsymbol{A}\boldsymbol{B}}, \dots, \overline{\boldsymbol{C}\boldsymbol{A}^{L-1}\boldsymbol{B}}\right)$, resulting in a complexity of $\mathcal{O}(Lh)$.

**DSS**  Instead of employ bilinear discretization, DSS [28] uses the Zero-Order Hold (ZOH) discretization:

$$\overline{\boldsymbol{A}} = \exp(\boldsymbol{A}\Delta t), \quad \overline{\boldsymbol{B}} = (\overline{\boldsymbol{A}} - \boldsymbol{I})(\Delta_t \boldsymbol{A})^{-1}\Delta_t \boldsymbol{B}, \quad \overline{\boldsymbol{C}} = \boldsymbol{C} \tag{6}$$

Based on this, they diagonalize $\overline{\boldsymbol{A}}$ to instantiate the SSM parameters:

$$\overline{\boldsymbol{A}} = \mathrm{diag}\left(\exp(\lambda_1 \Delta t), \exp(\lambda_2 \Delta t), \dots, \exp(\lambda_h \Delta t)\right), \quad \overline{\boldsymbol{B}} = \left(\frac{\exp(\lambda_i \Delta t) - 1}{\lambda_i \exp(L\lambda_i \Delta t) - 1}\right)_{1 \le i \le h}, \tag{7}$$

where $\exp \lambda_i$ is the $i$-th diagonal element of $\overline{\boldsymbol{A}}$.

**S5**  S5 [27] still employs ZOH discretization, but with a distinction, they utilize a MIMO setup, meaning all hidden dimensions of each token are jointly involved in state transition as in eq.(1) and (2) and employs parallel scanning for computation. To ensure efficiency, they set $\boldsymbol{A} = \boldsymbol{V}^{-1}\Lambda\boldsymbol{V}$ for diagonal reparameterization:

$$\tilde{\boldsymbol{A}} = \Lambda, \quad \tilde{\boldsymbol{x}}(t) = \boldsymbol{V}^{-1}x(t), \quad \boldsymbol{B} = \boldsymbol{V}^{-1}\boldsymbol{B}, \quad \tilde{\boldsymbol{C}} = \boldsymbol{C}\boldsymbol{V}, \tag{8}$$

this reduces the complexity of parallel scanning from $\mathcal{O}(h^3 L)$ to $\mathcal{O}(hL)$.

**Mamba (S6)**  Similar to DSS and S5, Mamba [20] utilizes ZOH for discretization and performs diagonal instantiation of $\boldsymbol{A}$, with computations carried out through parallel scanning. It is noteworthy that Mamba sets adaptive $\boldsymbol{B}_k(\boldsymbol{u}_k), \boldsymbol{C}_k(\boldsymbol{u}_k)$ and $\Delta t_k(\boldsymbol{u}_k), 1 \le k \le L$ for each step through learnable linear transformation (meaning that they adpoted a data-dependent SSM parameters setting), while still maintaining a SISO configuration.

**Mega**  Mega's SSM component is implemented in the form of Exponential Moving Average (EMA) [19]. Specifically, their Multi-dimensional Damped EMA is given by:

$$\boldsymbol{x}_k^{(i)} = \boldsymbol{\alpha}_i \odot \tilde{\boldsymbol{u}}_k^{(i)} + (1 - \boldsymbol{\alpha}_i \odot \boldsymbol{\delta}_i) \odot \boldsymbol{x}_{k-1}^{(i)}, \quad \tilde{\boldsymbol{u}}_k^{(i)} = \boldsymbol{\beta}_i \boldsymbol{u}_k^{(i)}, \quad \boldsymbol{y}_k^{(i)} = \boldsymbol{\eta}_i^\top \boldsymbol{x}_k^{(i)}, \tag{9}$$

where the expansion parameters $\boldsymbol{\beta}_i \in \mathbb{R}^h$, projection parameters $\boldsymbol{\eta}_i \in \mathbb{R}^h$, $\boldsymbol{\alpha}_i, \boldsymbol{\delta}_i \in \mathbb{R}^h$ represent the weighting and damping factors, respectively. This can be formalized equivalently as a SISO SSM with parameters $\boldsymbol{A}_i = \mathrm{diag}(1 - \boldsymbol{\alpha}_i \odot \boldsymbol{\delta}_i)$, $\boldsymbol{B}_i = \boldsymbol{\alpha}_i \odot \boldsymbol{\beta}_i$ and $\boldsymbol{C} = \boldsymbol{\eta}_i^\top$. Following the execution of SSM, Mega introduces Attention mechanism between the SSM output $\boldsymbol{y}$ and input sequence $\boldsymbol{u}$, which made it the most competitive Attention-based SSM structure [27].

Table 1: Comparisons among the test accuracy (%) of different SSM structures with different Training types on MNIST and CIFAR-10 test set. 'Best' and 'Last' mean the test performance on the best and last checkpoint, respectively. 'Diff' denotes the accuracy gap between the 'Best' and 'Last'. The best checkpoint is selected based on the highest RA on the test set under PGD-10.

| Dataset | Structure | Training Type | Clean | | | PGD-10 | | | AA | | |
|---|---|---|---|---|---|---|---|---|---|---|---|
| | | | Best | Last | Diff | Best | Last | Diff | Best | Last | Diff |
| **MNIST** | S4 | ST | 99.16 | 99.15 | 0.01 | 4.57 | 0.72 | 3.85 | 0.07 | 0.00 | 0.07 |
| | | PGD-AT | 53.26 | 50.11 | 3.15 | 99.92 | 99.87 | 0.05 | 0.26 | 0.00 | 0.26 |
| | | TRADES | 99.29 | 99.11 | 0.18 | 99.11 | 98.82 | 0.29 | 89.01 | 88.35 | 0.66 |
| | | FreeAT | 99.23 | 99.21 | 0.02 | 24.80 | 24.37 | 0.44 | 0.04 | 0.00 | 0.04 |
| | | YOPO | 11.35 | 11.35 | 0.00 | 11.35 | 11.35 | 0.00 | 11.35 | 11.35 | 0.00 |
| | DSS | ST | 99.38 | 99.33 | 0.05 | 7.34 | 2.72 | 4.62 | 0.23 | 0.00 | 0.23 |
| | | PGD-AT | 59.87 | 32.22 | 27.65 | 99.95 | 99.89 | 0.06 | 0.62 | 0.00 | 0.62 |
| | | TRADES | 99.21 | 99.16 | 0.05 | 98.97 | 98.75 | 0.22 | 89.46 | 88.20 | 1.26 |
| | | FreeAT | 99.25 | 99.23 | 0.02 | 16.78 | 13.90 | 2.88 | 0.04 | 0.00 | 0.04 |
| | | YOPO | 11.35 | 11.35 | 0.00 | 11.35 | 11.35 | 0.00 | 11.35 | 11.35 | 0.00 |
| | S5 | ST | 98.98 | 98.95 | 0.03 | 3.66 | 0.44 | 3.22 | 0.29 | 0.00 | 0.29 |
| | | PGD-AT | 96.95 | 20.20 | 76.75 | 99.92 | 99.86 | 0.06 | 0.65 | 0.00 | 0.65 |
| | | TRADES | 98.95 | 98.89 | 0.06 | 98.17 | 97.97 | 0.20 | 81.15 | 80.20 | 0.95 |
| | | FreeAT | 99.33 | 99.31 | 0.02 | 21.39 | 20.58 | 0.51 | 0.06 | 0.00 | 0.06 |
| | | YOPO | 11.35 | 11.35 | 0.00 | 11.35 | 11.35 | 0.00 | 11.35 | 11.35 | 0.00 |
| | Mega | ST | 99.34 | 99.30 | 0.04 | 9.54 | 7.93 | 1.61 | 0.25 | 0.00 | 0.25 |
| | | PGD-AT | 42.13 | 24.74 | 17.39 | 99.95 | 99.86 | 0.09 | 0.79 | 0.00 | 0.79 |
| | | TRADES | 99.24 | 99.20 | 0.04 | 99.05 | 98.97 | 0.08 | 11.09 | 10.90 | 0.19 |
| | | FreeAT | 99.46 | 99.43 | 0.03 | 24.69 | 13.54 | 11.15 | 0.13 | 0.00 | 0.13 |
| | | YOPO | 11.35 | 11.35 | 0.00 | 11.35 | 11.35 | 0.00 | 11.35 | 11.35 | 0.00 |
| | Mamba | ST | 98.93 | 98.86 | 0.07 | 25.53 | 19.35 | 6.18 | 0.09 | 0.00 | 0.09 |
| | | PGD-AT | 37.18 | 9.76 | 27.42 | 99.87 | 99.81 | 0.06 | 0.53 | 0.00 | 0.53 |
| | | TRADES | 98.84 | 98.83 | 0.01 | 98.86 | 98.79 | 0.07 | 53.03 | 52.85 | 0.18 |
| | | FreeAT | 99.05 | 99.03 | 0.02 | 35.69 | 30.20 | 15.49 | 0.26 | 0.00 | 0.00 |
| | | YOPO | 11.35 | 11.35 | 0.00 | 11.35 | 11.35 | 0.00 | 11.35 | 11.35 | 0.00 |
| **CIFAR-10** | S4 | ST | 78.78 | 78.58 | 0.20 | 10.38 | 0.00 | 10.38 | 0.96 | 0.00 | 0.96 |
| | | PGD-AT | 64.67 | 64.47 | 0.20 | 36.19 | 35.66 | 0.53 | 30.90 | 30.60 | 0.30 |
| | | TRADES | 63.91 | 63.78 | 0.13 | 36.00 | 35.42 | 0.58 | 30.55 | 30.55 | 0.00 |
| | | FreeAT | 69.69 | 69.64 | 0.05 | 20.91 | 19.63 | 1.28 | 15.57 | 15.29 | 0.28 |
| | | YOPO | 61.46 | 61.34 | 0.12 | 30.64 | 30.11 | 0.53 | 25.36 | 24.94 | 0.42 |
| | DSS | ST | 76.03 | 75.86 | 0.17 | 11.42 | 0.00 | 11.42 | 0.62 | 0.00 | 0.62 |
| | | PGD-AT | 64.92 | 64.70 | 0.22 | 37.31 | 37.07 | 0.24 | 31.65 | 31.60 | 0.05 |
| | | TRADES | 65.08 | 64.99 | 0.09 | 37.44 | 36.99 | 0.45 | 32.55 | 32.15 | 0.40 |
| | | FreeAT | 67.46 | 67.34 | 0.12 | 20.38 | 18.75 | 1.63 | 15.13 | 14.98 | 0.15 |
| | | YOPO | 59.87 | 57.77 | 1.10 | 31.77 | 30.89 | 0.88 | 26.36 | 26.01 | 0.35 |
| | S5 | ST | 70.32 | 70.30 | 0.02 | 7.62 | 0.00 | 7.62 | 0.48 | 0.00 | 0.48 |
| | | PGD-AT | 53.93 | 53.68 | 0.25 | 31.49 | 31.13 | 0.36 | 28.89 | 28.69 | 0.20 |
| | | TRADES | 57.68 | 57.23 | 0.45 | 31.34 | 31.10 | 0.24 | 26.56 | 25.73 | 0.83 |
| | | FreeAT | 67.15 | 67.00 | 0.15 | 19.59 | 18.56 | 1.03 | 13.90 | 13.67 | 0.23 |
| | | YOPO | 59.15 | 58.79 | 0.36 | 30.93 | 28.83 | 2.10 | 25.29 | 25.22 | 0.07 |
| | Mega | ST | 79.08 | 79.03 | 0.05 | 3.86 | 0.07 | 3.79 | 0.84 | 0.00 | 0.84 |
| | | PGD-AT | 72.54 | 72.08 | 0.46 | 40.53 | 30.68 | 9.85 | 25.79 | 25.26 | 0.53 |
| | | TRADES | 71.01 | 70.84 | 0.17 | 43.65 | 42.63 | 1.02 | 37.50 | 36.97 | 0.53 |
| | | FreeAT | 76.81 | 76.10 | 0.71 | 27.11 | 23.93 | 3.18 | 17.29 | 16.99 | 0.30 |
| | | YOPO | 74.37 | 72.98 | 1.39 | 36.39 | 31.56 | 4.83 | 28.68 | 27.13 | 1.55 |
| | Mamba | ST | 72.59 | 72.31 | 0.28 | 5.76 | 0.48 | 5.28 | 0.43 | 0.00 | 0.43 |
| | | PGD-AT | 58.69 | 58.54 | 0.15 | 35.98 | 35.07 | 0.91 | 33.07 | 32.28 | 0.79 |
| | | TRADES | 60.93 | 60.66 | 0.27 | 34.84 | 32.75 | 2.09 | 29.87 | 29.07 | 0.80 |
| | | FreeAT | 67.37 | 67.24 | 0.13 | 17.40 | 10.97 | 6.43 | 11.84 | 9.65 | 2.19 |
| | | YOPO | 65.31 | 63.36 | 1.95 | 28.08 | 27.61 | 0.47 | 23.63 | 22.85 | 0.78 |

## 3 Empirical Evaluation: Component Contributions to AT Gains

In this section, we empirically assess the natural AR of various SSM structures and their AR post AT to explore the AR enhancement provided by mainstream AT frameworks for SSMs and the generalization behavior of SSM variants under AT.

**Training Setup.** We adopt the ST and two most commonly used AT frameworks, PGD-AT [16] and TRADES [21], as well as two more efficient and advanced adversarial training frameworks, FreeAT [30] and YOPO [31], to conduct experiments on the MNIST, CIFAR-10, and Tiny-ImageNet datasets. For AT, we utilize a 10-step $\ell_\infty$ PGD (PGD-10) as the attack method. Following [21], on MNIST, we set the adversarial budget to $\|\epsilon\|_\infty = 0.3$, attack step size $\alpha = 0.04$, the KL divergence regularizer coefficient for TRADES $\beta = 1.0$, and the training epoch to 100. On CIFAR-10 and Tiny-Imagenet, we set $\|\epsilon\|_\infty = 0.031$, $\alpha = 0.007$, $\beta = 6.0$, and the training epoch to 180. After each training epoch of AT, we conduct adversarial testing on both the training and test sets to evaluate robustness and measure generalization on adversarial examples (i.e. Robust Generalization, RG) [21, 24, 32]. For the model architecture of SSMs, we set all models to a uniform 2-layer architecture with a hidden dimension $d = 128$ on MNIST, and a 4-layer architecture with $d = 128$ on CIFAR-10 and Tiny-Imagenet. Considering that SSMs are a class of models highly reliant on the sequential dependence, we adopt a sequence image classification setup to thoroughly investigate the AR of SSMs. For MNIST, we resize inputs to $(784, 1)$, for CIFAR-10 and Tiny-Imagenet, we resize inputs to $(1024, 3)$ and $(4096, 3)$. Further experimental results, including the results for Tiny-ImageNet, and detailed configurations are presented in the Appendix C and D.

**Evaluation Setting**. We record the CA and adversarial test accuracy (i.e. Robust Accuracy, RA) [10, 33] of each model at their best and final checkpoints across three training schemes to evaluate robustness and examine the robustness-generalization trade-off as well as the issue of RO. The

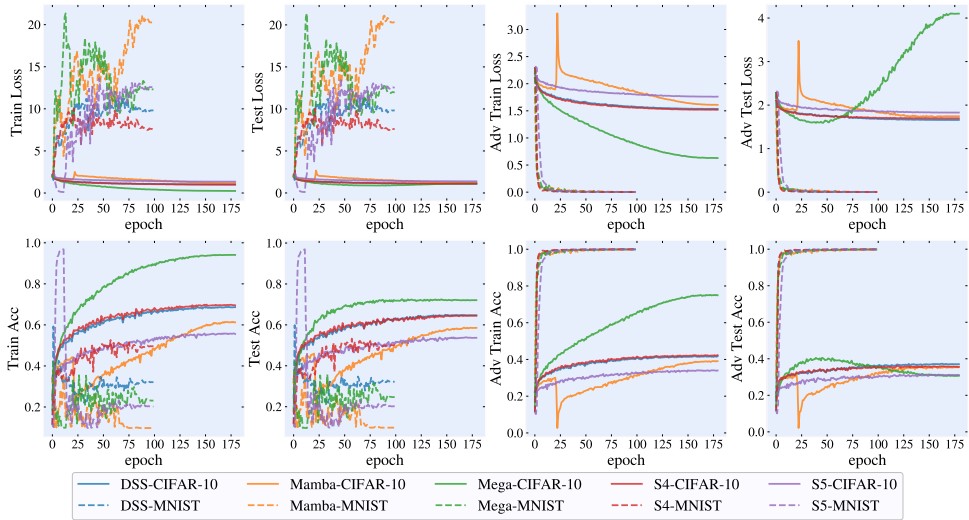

Figure 1: The PGD-AT training process, testing process, and the adversarial PGD-10 testing process on the training and testing datasets on CIFAR-10 and MNIST datasets.

results are presented in Table 1. Beyond adversarial testing with PGD-10, we incorporate AutoAttack (AA) [34] for a rigorous robustness evaluation. AA integrates four attack strategies: APGD-CE [34], APGD-DLR [34], FAB [35], and Square [36], serving as a powerful adversarial attack baseline widely used in AR evaluation [25, 33].

For SSMs trained with ST, almost all models exhibit no adversarial robustness, and the adversarial test loss for all models shows a continuous increase during the training process, as depicted in Fig. 5 in Appendix D, consistent with the conclusions drawn for convolutional networks [16]. However, it is noteworthy that the RA of Mega and Mamba trained standardly are higher than the rest of the SSM structures, especially on MNIST where this performance is particularly pronounced. This suggests that the flexibility introduced by Attention [19] and data-dependent SSM structures also helps the models to some extent in adapting to adversarial examples. Compared to the ST scenario, we are more interested in understanding the AR improvement of AT for SSMs and the dynamic behavior of SSMs during AT. To this end, we plot the training, testing, AT, and adversarial test loss and accuracy changes of various SSM variants on both datasets for PGD-AT (Fig. 1) and TRADES (Fig. 2). In conjunction with Table 1, we have the following observations:

**1) AT remains an effective strategy for enhancing the AR of SSMs, yet different SSM structures exhibit a clear trade-off between robustness and generalization.** All models benefit from AT, achieving higher RA with both PGD-AT and TRADES frameworks. Some intriguing phenomenons occurs on the MNIST dataset, where models trained under the PGD-AT framework exhibit a marked decrease in CA and almost no RA under AA. In contrast, TRADES does not lead to a significant reduction in CA and substantially improves robust accuracy under AA. This disparity is not observed on CIFAR-10 and Tiny-Imagenet. The phenomenon can be elucidated by findings from [37], which indicate that MNIST's limited semantic information may lead PGD-AT to fit to spurious rather than causal features, resulting in overfitting to $\ell_\infty$ PGD attacks. On the other hand, TRADES explicitly constrains the KL divergence between clean and adversarial sample logits, ensuring that the model learns causal features from clean data while adapting to adversarial distributions. Besides, on the MNIST dataset, all model architectures fail to converge when trained with YOPO. This advanced adversarial training strategy relies on the parameters of the model's input layer for auxiliary calculations, making it difficult to simply extend and be effective on non-convolutional structures. However, on the CIFAR-10 dataset, a noticeable decrease in CA is observed across all SSM structures after AT, particularly for S5, which experienced a $16.62\%$ drop in CA under PGD-AT, with other structures also showing approximately a $10\%$ reduction in CA, a similar phenomenon is observed on Tiny-ImageNet. This indicates a clear trade-off between robustness and generalization for SSM structures.

**2) The incorporation of Attention indeed yields a better trade-off between robustness and generalization, yet it also introduces a significant issue of RO.** Compared to other SSM structures, Mega exhibits the least CA degradation after AT, while compared to DSS, the model with the sec-

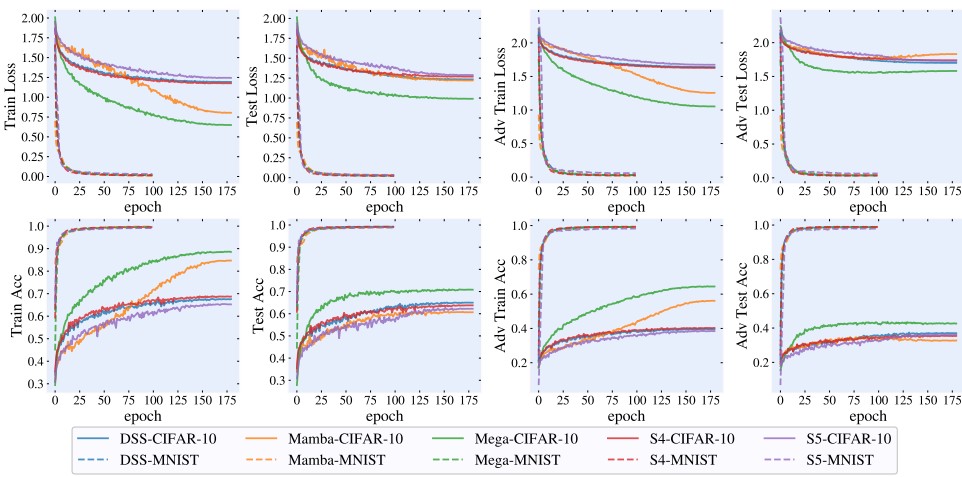

Figure 2: The TRADES training process, testing process, and the adversarial PGD-10 testing process on the training and testing datasets on CIFAR-10 and MNIST datasets.

ond least CA degradation, Mega achieves improvements of $4.21\%$ and $2.68\%$ in CA degradation under PGD-AT and TRADES, respectively. Moreover, Mega attains the highest adversarial accuracy against PGD-10 and AA attacks post-AT (in the best sense). This indicates that the introduction of Attention can indeed lead to a better robustness-generalization trade-off. However, from Fig. 1 and 2, it is observed that on CIFAR-10, both under PGD-AT and TRADES, Mega demonstrates a continuous decrease in AT loss and an increase in AT accuracy, but the RA in adversarial testing shows almost no growth trend in the middle and late stages of training, resulting in a difference of over $40\%$ (PGD-AT) and over $20\%$ (TRADES) between AT and test accuracy at the end of training. Particularly for PGD-AT, after achieving the best RA of $40.53\%$ at the 39th epoch, Mega's RA begins to gradually decline, eventually falling to $30.68\%$, showing a very evident RO issue. Additionally, Mamba also exhibits a clear RO issue during the TRADE training process, as depicted in Fig. 2, resulting in a difference of over $20\%$ between AT and test accuracy.

**3) The pure SSM structure exhibits a significant disadvantage in AT, and the data-dependent parametrization does not aid in the AT of SSMs.** S4, DSS and Mamba, due to their SISO characteristics, require the addition of linear layers after SSM calculation to aid in the interaction between different feature elements. While S5 that expanded with MIMO capabilities can be equivalent to a linear combination of multiple SISO SSMs, thus obviating the need for extra position-wise linear layers [27]. This makes the S5 block consist solely of a MIMO SSM. Nonetheless, on CIFAR-10, S5 demonstrates a marked disadvantage in RA and a worse robustness-generalization trade-off compared to other SSMs after training with both PGD-AT and TRADES (Tab. 1). Furthermore, we also observe that, despite the inclusion of linear layers, Mamba still exhibits a reduction in CA and RA compared to S4 and DSS, indicating that the extension of the data-dependent SSM does not contribute to the model's AT.

**Based on the observations, we give an answer to the questions Q1 and Q2**. SSMs alone seem to struggle to perform well in AT, while SSMs integrated with linear layers or Attention demonstrate a significant enhancement in AR compared to the pure SSM structure. However, SSMs with Attention, although showing the greatest gain in both robust and clean accuracy, exhibit a much more pronounced RO issue compared to SSMs with only linear layers integrated.

Based on these findings, we consider the following: Does the inherent nature of SSMs limit their benefit in AT? Is it the Attention mechanism itself causing overfitting to AT data? Can formal analysis of Attention's behavior in AT provide insights for finding ways to improve SSMs' AR while avoiding RO issues? We will delve deeper into these questions in the following section.

## 4    Component-wise Attribution: Theoretical and Experimental Analysis

In this section, we first theoretically investigate the stability boundaries of various SSMs under APs to understand why SSMs alone do not benefit significantly from AT. This theoretical analysis aims to uncover the reasons behind the limited effectiveness of AT on pure SSM structures. Next, we

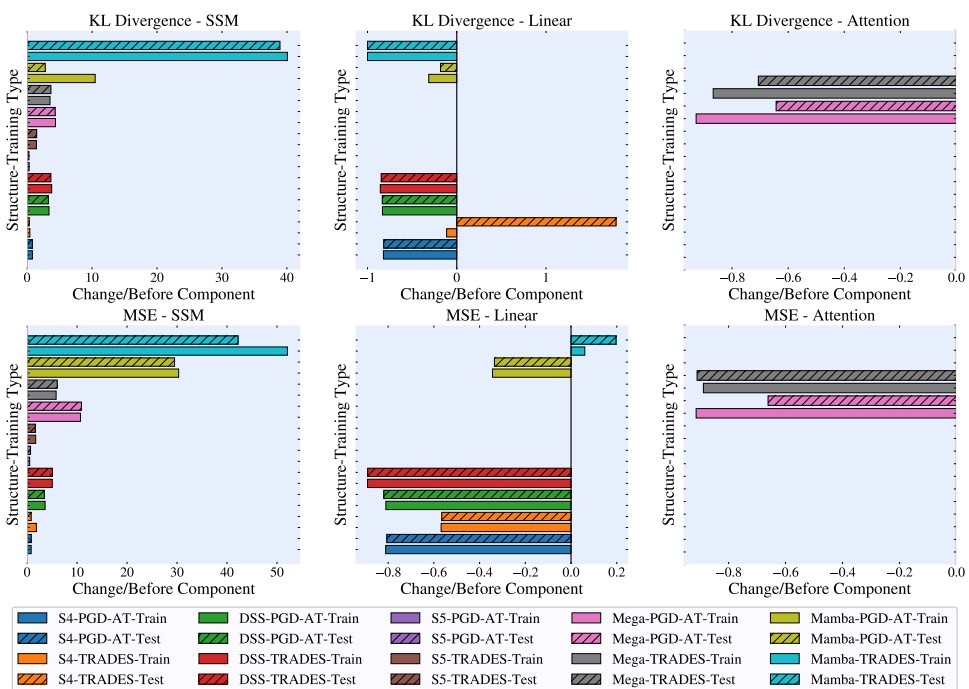

Figure 3: Changes in KL divergence and MSE before and after different components in various SSM structures are presented. The change for each component is calculated as: after component - before component. The data represents the change rate, calculated as: change / before component. Blank sections indicate the absence of a corresponding component. Bars with diagonal hatching represent results on the test set, while bars without hatching represent results on the training set.

experimentally validate each component integrated with SSMs to further elucidate their roles in enhancing AT performance. Finally, we conduct a formal analysis of why incorporating attention mechanisms aids in AT, aiming to provide insights for improving the robustness and performance of SSMs during AT.

## 4.1 Theoretical Analysis of SSM Stability Under APs

We consider a generalized scenario where the SSM input is denoted as $u = (u_1, u_2, \cdots, u_L)^T \in \mathbb{R}^{L \times 1}$, and the input after AP is $u' = (u'_1, u'_2, \cdots, u'_L)^T \in \mathbb{R}^{L \times 1}$. Let $\varepsilon = u' - u = (\varepsilon_1, \varepsilon_2, \cdots, \varepsilon_L)^T \in \mathbb{R}^{L \times 1}$, and assume $\mathbb{E}[\varepsilon] = \mu$, $\mathbb{E}[\varepsilon_k^2] \leq M$, for $k = 1, 2, \ldots, L$ (such assumptions are reasonable due to the perturbation budget constraints inherent to AP). Considering the generalized SSM setup:

$$y_k = \overline{C}_k \prod_{i=1}^{k-1} \overline{A_i B_k} u_1 + \cdots + \overline{C_2 A_1 B_2} u_{k-1} + \overline{C_1 B_1} u_k. \quad (k = 1, 2, \ldots, L). \tag{10}$$

The form in eq. (10) encompasses both the fixed-parameterized and the data-dependent scenario, where when $A_i = A, B_i = B, C_i = C$ ($i = 1, \ldots$), Equation (10) represents a fixed-parameterized SISO SSM. It is reasonable to consider only the SISO case here, as after diagonal reparameterization, each hidden dimension of the state transition equation in S5 can still be regarded as an independent SISO system. Based on the above settings, we have the following theorem:

**Theorem 4.1.1** *Given the SSM formalized as in eq. (10), the output error before and after perturbation, $\mathbb{E}_\varepsilon \left[ \|y' - y\|^2 \right]$, has the following upper and lower bounds:*

$$\mu^2 c_1 \sum_{j=1}^{L} \left[ \prod_{i=1}^{j-1} |\bar{\lambda}_i^{\min}| \right] (\overline{C}_j \overline{B}_j)^2 \leq \mathbb{E}_\varepsilon \left[ \|y' - y\|_2^2 \right] \leq c_2 M \sum_{i=1}^{L} \left[ \prod_{i=1}^{j-1} |\bar{\lambda}_i^{\max}| \right] (\overline{C}_j \overline{B}_j)^2, \tag{11}$$

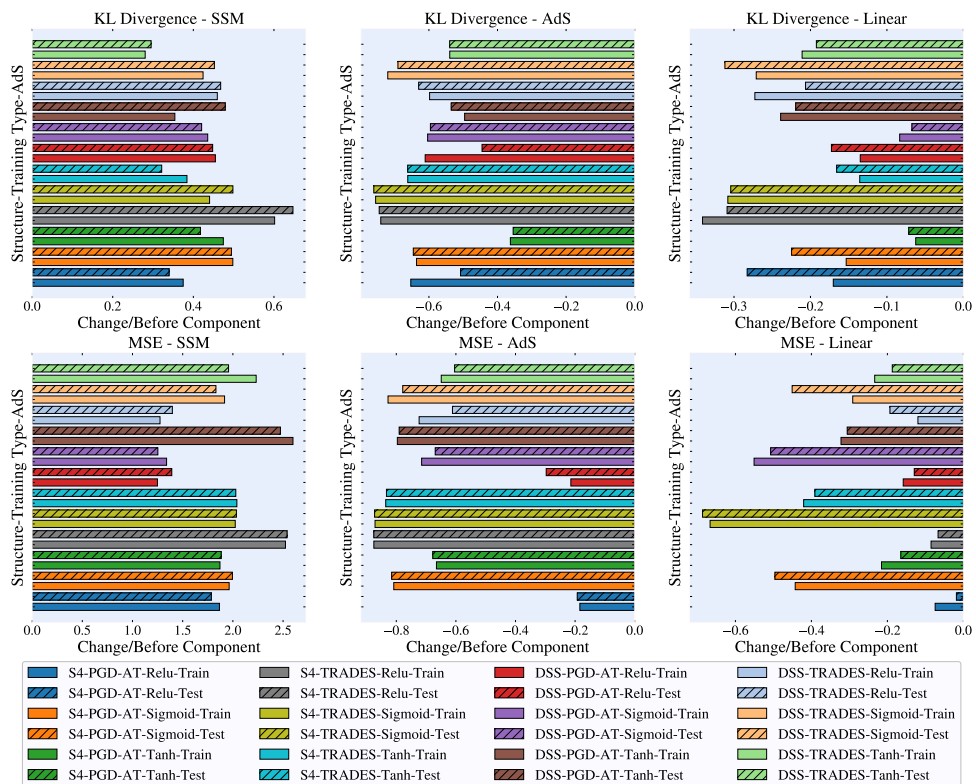

Figure 4: Changes in KL divergence and MSE before and after different components in the S4 and DSS with different AdS under PGD-AT and TRADES training on CIFAR-10. The change of the component is calculated as: after component - before component. The data in the figure represents the change rate which calculated as: change / before component. Bars with diagonal hatching represent the results on the test set, while bars without hatching represent that on the training set.

where $L$ denotes the length of the input sequence, $0 < c_1 \leq c_2$ are constants, and $\bar{\lambda}_i^{\max}$ and $\bar{\lambda}_i^{\min}$ are the eigenvalues of matrix $\overline{A}_i$ with the maximum and minimum absolute values, respectively.

Theorem 4.1.1 indicates that the output error of SSMs in the presence of AP is strictly related to the parameters of the SSMs. It should be noted that under the setting of fixed-parameterized SSMs (S4, DSS, S5, and Mega), the output error $\mathbb{E}_\varepsilon \left[ \|\boldsymbol{y}' - \boldsymbol{y}\|_2^2 \right]$ has a fixed lower bound $\mu^2 c_1 \sum_{j=1}^{L} \left[ \prod_{i=1}^{j-1} |\bar{\lambda}_i^{\min}| \right] (\overline{C}_j \overline{B}_j)^2$, which increases with the length of the sequence $L$.

**This implies that even when fixed-parameterized SSMs aim to stabilize the output of adversarial examples during AT by minimizing output error, they are constrained by a fixed lower bound on the error. This constraint leads to inevitable error accumulation, making it challenging for pure SSM structures to effectively benefit from AT.** While Mamba can avoid the issue of a fixed error lower bound through adaptive SSM parameterization, it may also introduce more severe negative impacts. Notice that during the ZOH discretization process, as $\Delta t_k \to 0$, the eigenvalues of $\Delta_t A$ in $\overline{B_k} = (\overline{A_k} - I)(\Delta t_k A_k)^{-1} \Delta t_k (\boldsymbol{u}_k) B$ tend towards 0. This results in an ill-conditioned problem when computing the inverse operation $(\Delta_t A)^{-1}$, causing $\|\overline{B}_k\| \to \infty$. Consequently, this leads to $\max_{\boldsymbol{u}} \overline{C}_k(\boldsymbol{u}_k)\overline{B}_k(\boldsymbol{u}_k)^2 \leq \|\overline{C}_k(\boldsymbol{u}_k)\|_2^2 \|\overline{B}_k(\boldsymbol{u}_k)\|_2^2 \to \infty$ in Mamba, potentially resulting in an almost infinite error bound. In contrast, training stable fixed-parameterized SSMs ensures a fixed upper bound on output error, thereby avoiding such error explosion issues.

## 4.2 Experimental Validation and Further Insights

To validate our theoretical analysis with quantitative metric and to deeply investigate the specific roles of components within various SSM structures during AT, we record the relative change

Table 2: Comparisons among the test accuracy (%) of S4 and DSS with different AdS modules and different AT types on CIFAR-10 and MNIST test set. 'Best' and 'Last' mean the test performance at the best and last epoch, respectively. 'Diff' denotes the accuracy gap between the 'Best' and 'Last', '–' indicates that the results of AdS are not used. The best checkpoint is selected based on the highest RA on the test set under PGD-10.

| Dataset | Structure | Training Type | AdS | Clean | | | PGD-10 | | | AA | | |
|---|---|---|---|---|---|---|---|---|---|---|---|---|
| | | | | Best | Last | Diff | Best | Last | Diff | Best | Last | Diff |
| MNIST | S4 | PGD-AT | – | 53.26 | 50.11 | 3.15 | 99.92 | 99.87 | 0.05 | 0.26 | 0.00 | 0.26 |
| | | | ReLU | 70.57 | 55.39 | 15.18 | 99.76 | 99.63 | 0.13 | 0.28 | 0.00 | 0.28 |
| | | | Sigmoid | 53.26 | 50.55 | 2.71 | 99.88 | 99.82 | 0.06 | 0.06 | 0.00 | 0.06 |
| | | | Tanh | 68.31 | 58.98 | 9.33 | 99.88 | 99.85 | 0.03 | 0.05 | 0.00 | 0.05 |
| | | TRADES | – | 99.29 | 99.11 | 0.18 | 99.11 | 98.82 | 0.29 | 89.01 | 88.35 | 0.66 |
| | | | ReLU | 99.37 | 99.33 | 0.04 | 99.22 | 99.21 | 0.01 | 90.13 | 90.05 | 0.08 |
| | | | Sigmoid | 99.28 | 99.27 | 0.01 | 99.08 | 99.05 | 0.03 | 89.79 | 89.60 | 0.19 |
| | | | Tanh | 99.33 | 99.28 | 0.05 | 99.25 | 99.17 | 0.08 | 90.17 | 90.00 | 0.17 |
| | | FreeAT | – | 99.23 | 99.21 | 0.02 | 24.80 | 24.37 | 0.44 | 0.04 | 0.00 | 0.04 |
| | | | ReLU | 99.28 | 99.23 | 0.05 | 25.76 | 25.23 | 0.53 | 0.07 | 0.00 | 0.07 |
| | | | Sigmoid | 99.17 | 99.15 | 0.02 | 25.32 | 25.18 | 0.16 | 0.04 | 0.00 | 0.04 |
| | | | Tanh | 99.30 | 99.21 | 0.09 | 25.53 | 25.02 | 0.51 | 0.04 | 0.00 | 0.04 |
| | | YOPO | – | 11.35 | 11.35 | 0.00 | 11.35 | 11.35 | 0.00 | 11.35 | 11.35 | 0.00 |
| | | | ReLU | 11.35 | 11.35 | 0.00 | 11.35 | 11.35 | 0.00 | 11.35 | 11.35 | 0.00 |
| | | | Sigmoid | 11.35 | 11.35 | 0.00 | 11.35 | 11.35 | 0.00 | 11.35 | 11.35 | 0.00 |
| | | | Tanh | 11.35 | 11.35 | 0.00 | 11.35 | 11.35 | 0.00 | 11.35 | 11.35 | 0.00 |
| | DSS | PGD-AT | – | 59.87 | 32.22 | 27.65 | 99.95 | 99.89 | 0.06 | 0.62 | 0.00 | 0.62 |
| | | | ReLU | 58.82 | 49.00 | 9.82 | 99.86 | 99.80 | 0.06 | 0.09 | 0.00 | 0.09 |
| | | | Sigmoid | 50.24 | 46.05 | 4.19 | 99.97 | 99.94 | 0.03 | 0.23 | 0.00 | 0.23 |
| | | | Tanh | 51.49 | 42.40 | 9.09 | 99.97 | 99.95 | 0.02 | 0.43 | 0.00 | 0.43 |
| | | TRADES | – | 99.21 | 99.16 | 0.05 | 98.97 | 98.75 | 0.22 | 89.46 | 88.20 | 1.26 |
| | | | ReLU | 99.30 | 99.27 | 0.03 | 99.00 | 98.99 | 0.01 | 88.96 | 88.80 | 0.16 |
| | | | Sigmoid | 99.33 | 99.30 | 0.03 | 99.08 | 99.08 | 0.00 | 89.25 | 89.00 | 0.25 |
| | | | Tanh | 99.30 | 99.27 | 0.03 | 99.00 | 98.95 | 0.05 | 90.34 | 90.25 | 0.09 |
| | | FreeAT | – | 99.25 | 99.23 | 0.02 | 16.78 | 13.90 | 2.88 | 0.04 | 0.00 | 0.04 |
| | | | ReLU | 99.45 | 99.43 | 0.02 | 21.78 | 19.80 | 1.98 | 0.06 | 0.00 | 0.06 |
| | | | Sigmoid | 99.24 | 99.19 | 0.05 | 21.47 | 19.33 | 2.14 | 0.07 | 0.00 | 0.07 |
| | | | Tanh | 99.36 | 99.32 | 0.04 | 21.67 | 19.72 | 1.95 | 0.04 | 0.00 | 0.04 |
| | | YOPO | – | 11.35 | 11.35 | 0.00 | 11.35 | 11.35 | 0.00 | 11.35 | 11.35 | 0.00 |
| | | | ReLU | 11.35 | 11.35 | 0.00 | 11.35 | 11.35 | 0.00 | 11.35 | 11.35 | 0.00 |
| | | | Sigmoid | 11.35 | 11.35 | 0.00 | 11.35 | 11.35 | 0.00 | 11.35 | 11.35 | 0.00 |
| | | | Tanh | 11.35 | 11.35 | 0.00 | 11.35 | 11.35 | 0.00 | 11.35 | 11.35 | 0.00 |
| CIFAR-10 | S4 | PGD-AT | – | 64.67 | 64.47 | 0.20 | 36.19 | 35.66 | 0.53 | 30.90 | 30.60 | 0.30 |
| | | | ReLU | 69.23 | 69.10 | 0.13 | 38.06 | 37.83 | 0.23 | 32.83 | 31.85 | 0.98 |
| | | | Sigmoid | 68.97 | 68.79 | 0.18 | 37.67 | 37.47 | 0.20 | 32.52 | 31.65 | 0.87 |
| | | | Tanh | 70.38 | 70.19 | 0.19 | 39.13 | 38.42 | 0.71 | 33.64 | 33.35 | 0.29 |
| | | TRADES | – | 63.91 | 63.78 | 0.13 | 36.00 | 35.42 | 0.58 | 30.55 | 30.55 | 0.00 |
| | | | ReLU | 68.65 | 68.42 | 0.23 | 40.22 | 39.67 | 0.55 | 35.92 | 35.35 | 0.57 |
| | | | Sigmoid | 67.71 | 67.55 | 0.16 | 38.90 | 38.46 | 0.44 | 34.69 | 33.95 | 0.74 |
| | | | Tanh | 66.96 | 66.49 | 0.47 | 38.11 | 37.98 | 0.13 | 33.96 | 33.35 | 0.61 |
| | | FreeAT | – | 69.69 | 69.64 | 0.05 | 20.91 | 19.63 | 1.28 | 15.57 | 15.29 | 0.28 |
| | | | ReLU | 70.56 | 70.47 | 0.09 | 21.64 | 20.52 | 1.12 | 16.34 | 16.15 | 0.19 |
| | | | Sigmoid | 70.21 | 70.17 | 0.04 | 21.39 | 20.24 | 1.15 | 16.17 | 16.01 | 0.16 |
| | | | Tanh | 70.47 | 70.38 | 0.09 | 21.48 | 20.28 | 1.20 | 16.25 | 16.12 | 0.13 |
| | | YOPO | – | 61.46 | 61.34 | 0.12 | 30.64 | 30.11 | 0.53 | 25.36 | 24.94 | 0.42 |
| | | | ReLU | 62.74 | 62.61 | 0.13 | 31.63 | 31.24 | 0.39 | 26.64 | 26.35 | 0.29 |
| | | | Sigmoid | 62.61 | 62.42 | 0.19 | 31.52 | 31.18 | 0.36 | 26.61 | 26.27 | 0.34 |
| | | | Tanh | 62.71 | 62.69 | 0.12 | 31.58 | 31.25 | 0.33 | 26.57 | 26.31 | 0.26 |
| | DSS | PGD-AT | – | 64.92 | 64.70 | 0.22 | 37.31 | 37.07 | 0.24 | 31.65 | 31.60 | 0.05 |
| | | | ReLU | 70.44 | 70.12 | 0.32 | 40.68 | 39.88 | 0.80 | 28.48 | 28.20 | 0.28 |
| | | | Sigmoid | 68.19 | 68.14 | 0.05 | 38.31 | 38.08 | 0.23 | 33.71 | 33.35 | 0.36 |
| | | | Tanh | 69.52 | 69.31 | 0.21 | 38.89 | 38.06 | 0.83 | 33.89 | 33.70 | 0.19 |
| | | TRADES | – | 65.08 | 64.99 | 0.09 | 37.44 | 36.99 | 0.45 | 32.55 | 32.15 | 0.40 |
| | | | ReLU | 69.39 | 69.09 | 0.30 | 41.24 | 41.04 | 0.20 | 37.56 | 36.50 | 1.06 |
| | | | Sigmoid | 67.66 | 67.46 | 0.20 | 39.94 | 39.72 | 0.22 | 35.19 | 35.15 | 0.04 |
| | | | Tanh | 69.22 | 68.93 | 0.29 | 40.89 | 40.67 | 0.22 | 36.78 | 36.65 | 0.13 |
| | | FreeAT | – | 67.46 | 67.34 | 0.12 | 20.38 | 18.75 | 1.63 | 15.13 | 14.98 | 0.15 |
| | | | ReLU | 68.28 | 68.08 | 0.20 | 21.18 | 20.37 | 0.81 | 15.89 | 15.68 | 0.21 |
| | | | Sigmoid | 68.07 | 67.89 | 0.08 | 21.05 | 20.21 | 0.84 | 15.78 | 15.62 | 0.16 |
| | | | Tanh | 68.30 | 68.07 | 0.23 | 21.13 | 20.41 | 0.72 | 15.83 | 15.61 | 0.22 |
| | | YOPO | – | 59.87 | 57.77 | 1.10 | 31.77 | 30.89 | 0.88 | 26.36 | 26.01 | 0.35 |
| | | | ReLU | 60.33 | 59.39 | 0.94 | 32.53 | 32.39 | 0.24 | 27.18 | 27.01 | 0.17 |
| | | | Sigmoid | 60.24 | 59.40 | 0.84 | 32.38 | 32.20 | 0.18 | 27.01 | 26.87 | 0.14 |
| | | | Tanh | 60.31 | 59.34 | 0.97 | 32.47 | 32.26 | 0.21 | 27.23 | 27.05 | 0.18 |

in feature sequence differences $d(\boldsymbol{f}', \boldsymbol{f})/\|\boldsymbol{f}\|$ (%) on CIFAR-10 for each model post-AT, where $\boldsymbol{f}', \boldsymbol{f} \in \mathbb{R}^{L \times d}$ represent the feature sequences corresponding to adversarial and clean inputs, respectively. We measure the differences using two metrics: the Mean Squared Error (MSE) $\frac{1}{T}\|\boldsymbol{f}' - \boldsymbol{f}\|_2^2$, to quantify the absolute discrepancy between $f$ and $f'$, and the Kullback-Leibler (KL) divergence [38] $\mathbf{KL}(\mathcal{S}(\boldsymbol{f})\|\mathcal{S}(\boldsymbol{f}'))$, to evaluate the distributional divergence between $f'$ and $f$, where $\mathcal{S}$ represents the Softmax operation over the sequence dimension, given by $\mathcal{S}(f)_{[k,j]} = \exp(f)_{[k,j]}/\sum_{i=1}^{T}\exp(f)_{[i,j]}$. We average the statistical results from both the training and test sets, analyzing the performance differences between them to attribute the issue of RO. For each component, we calculate the average results across all layers. As shown in Fig. 3, there is an increase in both MSE and KL divergence before and after the SSM in all models. Notably, Mamba exhibits an over 40 times increase in KL and MSE on both the training and test sets after training with TRADES, significantly higher than the changes observed in other fixed-parameterized SSMs. These findings are consistent with our theoretical analysis, indicating that SSMs inherently struggle to reduce output discrepancies through AT, while data-dependent SSMs may experience an explosion of output errors during AT.

It is noteworthy that both the linear layers and the Attention mechanism exhibit a significant reduction in MSE and KL divergence, implying that they indeed serve to diminish the discrepancy between feature sequences of clean and adversarial input during AT. This explains why SSMs integrated with Attention and linear layers achieve higher CA and Robust RA in AT. Compared to linear layers, Attention induces a more pronounced reduction in MSE and KL divergence. However, while Attention after PGD-AT achieves more than a 90% decrease in MSE and KL divergence on the training data, it only yields about a 60% reduction on the test set, which is significantly weaker than the improvements observed on the training set. In contrast, other components do not exhibit such a notable degradation in difference reduction on the test set, indicating that **the RO issue essentially stems from Attention itself, rather than other components**. In fact, the incorporation of Attention introduces excessive model complexity, which, according to the conclusions in previous work [39], also limits the model's RG. According to eq. (11), the introduction of Attention effectively allows the model to adaptively scale the output error: $\mathbb{E}\left[\|\mathcal{A}(\boldsymbol{y}')\boldsymbol{y} - \mathcal{A}(\boldsymbol{y})\boldsymbol{y}\|_2^2\right]$, where $\mathcal{A}$ denotes the Self-Attention operation. This inspires us to consider: *whether incorporating an Ads to the SSM's output sequence $\boldsymbol{y}$ with minimal added complexity could enable fixed-parameterized SSMs to approach the AT performance of Attention while avoiding the introduction of RO?*

To answer this question, we introduced an AdS module after the SSM of S4 and DSS, to adaptively scale the SSM output $\boldsymbol{y}$: $\boldsymbol{y}_k = \sigma(\mathbb{L}_1(\boldsymbol{y}_k)) \odot \boldsymbol{y}_k + \sigma(\mathbb{L}_2(\boldsymbol{y}_k))$, where $\mathbb{L}_1$ and $\mathbb{L}_2$ are learnable linear mappings, $\mathbb{L}_2(\boldsymbol{y}_k)$ is used to assist in adaptive scaling to adjust the output bias, and $\sigma$ is an activation function, which includes ReLU, Tanh, or Sigmoid. We conduct AT on SSMs with AdS integrated with different activations on CIFAR-10 and MNIST using PGD-AT and TRADES, and record the best and final CA and RA. As shown in the results in Table 2, regardless of the type of activation integrated, S4 and DSS with AdS showed more than a 3% improvement in CA and over a 2% improvement in RA. In particular, the AdS with ReLU as activation brought a 4.1% improvement in CA and a 3.97% improvement in RA for DSS when trained with TRADES, nearly matching Mega's performance in terms of CA and RA under PGD-10 and AA, without exhibiting RO issues observed in Mega with PGD-AT. The experimental results on Tiny-ImageNet (Tab. 6) also support the same conclusion. To further confirm that AdS can indeed help reduce SSM output discrepancies, similar to the effect of Attention, we also quantify the differences in the intermediate layer feature sequences for each component of S4 and DSS after integrating AdS, as shown in Fig. 4. The feature sequence differences before and after AdS, in terms of both KL divergence and MSE, shows a decrease, indicating that the introduction of AdS does help in reducing the disparity between clean and adversarial feature sequences. This provides an affirmative answer to our question, suggesting that the primary advantage of Attention in enhancing SSM's performance in AT is due to its adaptive sequence scaling. Incorporating a low-complexity adaptive scaling module can improve the SSM model's CA and RA in AT while preventing RO issues.

## 5 Conclusion

In this study, we evaluate the performance of various SSMs structures in AT and investigate which components positively assist SSMs in this context. Our findings reveal the robustness-generalization trade-off inherent in SSMs during AT. Although the integration of Attention improves both the CA and RA of SSMs, it also increases model complexity, leading to RO issues. Through experimental and theoretical analyses, we demonstrate that pure SSM structures hardly benefit from AT, whereas Attention facilitates further reduction of output error through sequence scaling. Inspired by this, we show that introducing a simple and effective AdS module to SSMs effectively enhances their AT performance and prevents RO. This provides valuable guidance and insights for the future design of more robust SSM structures.

## 6 Acknowledgement

This work is supported by the National Science and Technology Major Project (2023ZD0121403). We extend our gratitude to the anonymous reviewers for their insightful feedback, which has greatly contributed to the improvement of this paper.

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

# A  Mathematical Derivations

## A.1  The Proof of Theorem 4.1.1

Denote

$$
\boldsymbol{H} \triangleq \begin{bmatrix}
\overline{C_1 B_1} & 0 & \cdots & 0 \\
\overline{C_2 A_1 B_2} & \overline{C_1 B_1} & \cdots & 0 \\
\vdots & \vdots & \ddots & \vdots \\
\overline{C_L}\left[\prod_{i=1}^{L-1}\overline{A_i}\right]\overline{B_L} & \overline{C_{L-1}}\left[\prod_{i=1}^{L-1}\overline{A_i}\right]\overline{B_{L-1}} & \cdots & \overline{C_1 B_1}
\end{bmatrix},
$$

then we have $\boldsymbol{y} = (y_1, y_2, \cdots, y_L) = \boldsymbol{H}u$, $\boldsymbol{y}' = (y'_1, y'_2, \cdots, y'_L) = \boldsymbol{H}(u + \varepsilon)$

(i) First, we explain how the lower bound is derived

$$
\begin{aligned}
\mathbb{E}\left[(\boldsymbol{y}' - \boldsymbol{y})^{\top}(\boldsymbol{y}' - \boldsymbol{y})\right] &= \mathbb{E}\left[(\boldsymbol{H}\varepsilon)^{\top}(\boldsymbol{H}\varepsilon)\right] \\
&\geq \left[\mathbb{E}\left[(\boldsymbol{y}' - \boldsymbol{y})\right]\right]^{\top}\left[\mathbb{E}\left[(\boldsymbol{y}' - \boldsymbol{y})\right]\right] \quad (Cauchy\ Schiwarz) \\
&= (\mu \boldsymbol{H}\boldsymbol{1})^T(\mu \boldsymbol{H}\boldsymbol{1}) \\
&= \mu^2 \boldsymbol{1}^T \boldsymbol{H}^T \boldsymbol{H}\boldsymbol{1},
\end{aligned}
\tag{12}
$$

where $\boldsymbol{1} = (1, 1, \cdots, 1)^T$. Denote $\boldsymbol{H} \triangleq [\boldsymbol{h}_1, \boldsymbol{h}_2, \cdots, \boldsymbol{h}_L]$, where $\boldsymbol{h}_1, \boldsymbol{h}_2, \cdots, \boldsymbol{h}_L$ are column vectors of $\boldsymbol{H}$, i.e. $\boldsymbol{h}_1 = \left(\overline{C_1 B_1}, \overline{C_2 A_1 B_2}, \cdots, \overline{C_L}\left[\prod_{i=1}^{L-1}\overline{A_i}\right]\overline{B_L}\right)$, $\boldsymbol{h}_2 = \left(0, \overline{C_1 B_1}, \cdots, \overline{C_{L-1}}\left[\prod_{i=1}^{L-2}\overline{A_i}\right]\overline{B_{L-1}}\right), \cdots, \boldsymbol{h}_L = \left(0, 0, \cdots, \overline{C_1 B_1}\right)$, then the inequality 12 can be written as :

$$
\begin{aligned}
\mathbb{E}\left[(\boldsymbol{y}' - \boldsymbol{y})^{\top}(\boldsymbol{y}' - \boldsymbol{y})\right] &\geq \mu^2 (\sum_{i=1}^{L}\boldsymbol{h}_i)^T(\sum_{i=1}^{L}\boldsymbol{h}_i) \\
&= \mu^2 \begin{bmatrix}
\overline{C_1 B_1} \\
\overline{C_2 A_1 B_2} + \overline{C_1 B_1} \\
\vdots \\
\sum_{j=1}^{L}\overline{C_j}\left[\prod_{i=1}^{j-1}\overline{A_i}\right]\overline{B_j}
\end{bmatrix}^T
\begin{bmatrix}
\overline{C_1 B_1} \\
\overline{C_2 A_1 B_2} + \overline{C_1 B_1} \\
\vdots \\
\sum_{j=1}^{L}\overline{C_j}\left[\prod_{i=1}^{j-1}\overline{A_i}\right]\overline{B_j}
\end{bmatrix} \\
&= \mu^2 \left(\boldsymbol{E}\begin{bmatrix}
\overline{C_1 B_1} \\
\overline{C_2 A_1 B_2} \\
\vdots \\
\overline{C_L}\left[\prod_{i=1}^{L-1}\overline{A_i}\right]\overline{B_L}
\end{bmatrix}\right)^T
\left(\boldsymbol{E}\begin{bmatrix}
\overline{C_1 B_1} \\
\overline{C_2 A_1 B_2} \\
\vdots \\
\overline{C_L}\left[\prod_{i=1}^{L-1}\overline{A_i}\right]\overline{B_L}
\end{bmatrix}\right) \\
&= \mu^2 \boldsymbol{h}_1^T \boldsymbol{E}^T \boldsymbol{E}\boldsymbol{h}_1.
\end{aligned}
\tag{13}
$$

where $\boldsymbol{E} = \begin{bmatrix}
1 & & & & & \\
1 & 1 & & & & \\
1 & 1 & 1 & & & \\
1 & 1 & 1 & 1 & & \\
\vdots & \vdots & \vdots & \vdots & \ddots & \\
1 & 1 & 1 & 1 & \cdots & 1
\end{bmatrix}$ is a lower triangular matrix with all elements being 1.

Therefore, $\boldsymbol{E}^T\boldsymbol{E}$ is a real symmetric positive definite matrix and is full rank, then there exists an orthogonal matrix $\boldsymbol{Q}$ and a diagonal matrix $\boldsymbol{\Lambda} = diag(\lambda_1, \lambda_2, \cdots, \lambda_L)$, where $\lambda_i$ are eigenvalues of $\boldsymbol{E}^T\boldsymbol{E}$, and $\lambda_i$ must satisfy $\lambda_i > 0$, denote $\lambda_{\max} = \max\{\lambda_1, \lambda_2, \ldots, \lambda_L\} = c_2$ and $\lambda_{\min} =$

$\min\{\lambda_1, \lambda_2, \ldots, \lambda_L\} = c_1$, we have,

$$
\begin{aligned}
\mathbb{E}\left[(\boldsymbol{y}' - \boldsymbol{y})^\top (\boldsymbol{y}' - \boldsymbol{y})\right] &\geq \mu^2 \boldsymbol{h}_1^T \boldsymbol{E}^T \boldsymbol{E} \boldsymbol{h}_1 \\
&= \mu^2 \boldsymbol{h}_1^T \boldsymbol{Q}^T \boldsymbol{\Lambda} \boldsymbol{Q} \boldsymbol{h}_1 \\
&\geq \mu^2 c_1 \boldsymbol{h}_1^T \boldsymbol{Q}^T \boldsymbol{I} \boldsymbol{Q} \boldsymbol{h}_1 \\
&= \mu^2 c_1 \boldsymbol{h}_1^T \boldsymbol{h}_1 \\
&= \mu^2 c_1 \sum_{j=1}^{L} (\overline{\boldsymbol{C}}_j \left[\prod_{i=1}^{j-1} \overline{\boldsymbol{A}_i}\right] \overline{\boldsymbol{B}}_j)^2.
\end{aligned}
\tag{14}
$$

(ii) Perform a similar derivation for the upper bound

$$
\begin{aligned}
\mathbb{E}\left[(\boldsymbol{y}' - \boldsymbol{y})^\top (\boldsymbol{y}' - \boldsymbol{y})\right] &= \mathbb{E}\left[(\boldsymbol{H}\boldsymbol{\varepsilon})^\top (\boldsymbol{H}\boldsymbol{\varepsilon})\right] \\
&= \mathbb{E}\left[\boldsymbol{\varepsilon}^T \boldsymbol{H}^T \boldsymbol{H} \boldsymbol{\varepsilon}\right] \\
&= \mathbb{E}\left[(\sum_{i=1}^{L} \varepsilon_i \boldsymbol{h}_i)^T (\sum_{i=1}^{L} \varepsilon_i \boldsymbol{h}_i)\right] \\
&= \mathbb{E}\left(\begin{bmatrix} \overline{\boldsymbol{C}_1 \boldsymbol{B}_1} \\ \overline{\boldsymbol{C}_2 \boldsymbol{A}_1 \boldsymbol{B}_2} \\ \vdots \\ \overline{\boldsymbol{C}_L} \left[\prod_{i=1}^{L-1} \overline{\boldsymbol{A}_i}\right] \overline{\boldsymbol{B}_L} \end{bmatrix}^T \boldsymbol{E}_\varepsilon^T \boldsymbol{E}_\varepsilon \begin{bmatrix} \overline{\boldsymbol{C}_1 \boldsymbol{B}_1} \\ \overline{\boldsymbol{C}_2 \boldsymbol{A}_1 \boldsymbol{B}_2} \\ \vdots \\ \overline{\boldsymbol{C}_L} \left[\prod_{i=1}^{L-1} \overline{\boldsymbol{A}_i}\right] \overline{\boldsymbol{B}_L} \end{bmatrix}\right) \\
&= \mathbb{E}[\boldsymbol{h}_1^T \boldsymbol{E}_\varepsilon^T \boldsymbol{E}_\varepsilon \boldsymbol{h}_1] \\
&= \mathbb{E}[\boldsymbol{h}_1^T \boldsymbol{\Lambda}_\varepsilon \boldsymbol{E}^T \boldsymbol{E} \boldsymbol{\Lambda}_\varepsilon \boldsymbol{h}_1] \\
&\leq c_2 \mathbb{E}[\boldsymbol{h}_1^T \boldsymbol{\Lambda}_\varepsilon \boldsymbol{Q}^T \boldsymbol{Q} \boldsymbol{\Lambda}_\varepsilon \boldsymbol{h}_1] \\
&= c_2 \mathbb{E}[\boldsymbol{h}_1^T \boldsymbol{\Lambda}_\varepsilon^2 \boldsymbol{h}_1] \\
&= c_2 \sum_{i=1}^{L} \mathbb{E}\varepsilon_i^2 (\overline{\boldsymbol{C}}_j \left[\prod_{i=1}^{j-1} \overline{\boldsymbol{A}_i}\right] \overline{\boldsymbol{B}}_j)^2 \\
&\leq c_2 M \sum_{i=1}^{L} (\overline{\boldsymbol{C}}_j \left[\prod_{i=1}^{j-1} \overline{\boldsymbol{A}_i}\right] \overline{\boldsymbol{B}}_j)^2,
\end{aligned}
\tag{15}
$$

where $\boldsymbol{E}_\varepsilon = \begin{bmatrix} \varepsilon_1 & & & \\ \varepsilon_1 & \varepsilon_2 & & \\ \vdots & \vdots & \ddots & \\ \varepsilon_1 & \varepsilon_2 & \cdots & \varepsilon_L \end{bmatrix} = \boldsymbol{E} \cdot diag(\varepsilon_1, \varepsilon_2, \cdots, \varepsilon_L) \triangleq \boldsymbol{E} \boldsymbol{\Lambda}_\varepsilon$. Let the eigenvalue with the maximum absolute value of matrix $\overline{\boldsymbol{A}_i}$ be denoted as $\bar{\lambda}_i^{\max}$ and the eigenvalue with the minimum absolute value as $\bar{\lambda}_i^{\min}$ note that $\prod_{i=1}^{j-1} |\bar{\lambda}_i^{\min}| (\overline{\boldsymbol{C}}_j \overline{\boldsymbol{B}}_j)^2 \leq (\overline{\boldsymbol{C}}_j \left[\prod_{i=1}^{j-1} \overline{\boldsymbol{A}_i}\right] \overline{\boldsymbol{B}}_j)^2 \leq \prod_{i=1}^{j-1} |\bar{\lambda}_i^{\max}| (\overline{\boldsymbol{C}}_j \overline{\boldsymbol{B}}_j)^2$, which means:

$$
\mu^2 c_1 \sum_{j=1}^{L} \left[\prod_{i=1}^{j-1} |\bar{\lambda}_i^{\min}|\right] (\overline{\boldsymbol{C}}_j \overline{\boldsymbol{B}}_j)^2 \leq \mathbb{E}\left[(\boldsymbol{y}' - \boldsymbol{y})^\top (\boldsymbol{y}' - \boldsymbol{y})\right] \leq c_2 M \sum_{i=1}^{L} \left[\prod_{i=1}^{j-1} |\bar{\lambda}_i^{\max}|\right] (\overline{\boldsymbol{C}}_j \overline{\boldsymbol{B}}_j)^2.
\tag{16}
$$

Table 3: Specific architecture of models with different SSM structures on each dataset.

| Structure | CIFAR-10 | | | MNIST | | |
| | Input | SSM Layer×4 | Output | Input | SSM Layer×2 | Output |
|---|---|---|---|---|---|---|
| S4 | Linear(3,128) LayerNorm(128) | S4 SSM GeLU Linear(128,128) GeLU LayerNorm(128) | Linear(128,10) | Linear(1,128) LayerNorm(128) | S4 SSM GeLU Linear(128,128) GeLU LayerNorm(128) | Linear(128,10) |
| DSS | Linear(3,128) LayerNorm(128) | DSS SSM GeLU Linear(128,128) GeLU LayerNorm(128) | Linear(128,10) | Linear(1,128) LayerNorm(128) | DSS SSM GeLU Linear(128,128) GeLU LayerNorm(128) | Linear(128,10) |
| S5 | Linear(3,128) LayerNorm(128) | S5 SSM LayerNorm(128) | Linear(128,10) | Linear(1,128) LayerNorm(128) | S5 SSM LayerNorm(128) | Linear(128,10) |
| Mega | Linear(3,128) LayerNorm(128) | EMA Attention Softmax LayerNorm(128) Linear(128, 256) SiLU Linear(256, 128) LayerNorm(128) | Linear(128,10) | Linear(1,128) LayerNorm(128) | EMA Attention Softmax LayerNorm(128) Linear(128, 32) SiLU Linear(32, 128) LayerNorm(128) | Linear(128,10) |
| Mamba | Linear(3,128) LayerNorm(128) | Mamba SSM Linear(256,128) LayerNorm(128) | Linear(128,10) | Linear(1,128) LayerNorm(128) | Mamba SSM Linear(256,128) LayerNorm(128) | Linear(128,10) |

Table 4: **Model** and **Training** parameters on MNIST, CIFAR-10 and Tiny-Imagenet datasets.

| | Parameters | MNIST | CIFAR-10 | Tiny-Imagenet |
|---|---|---|---|---|
| Model | Input Dim | 1 | 3 | 3 |
| | SSM Layers | 2 | 4 | 4 |
| | Model Dim | 128 | 128 | 128 |
| | State Dim | 32 | 64 | 64 |
| | Output Dim | 10 | 10 | 50 |
| | Reduction Before Head | Mean | Mean | Mean |
| Training | Optimizer | AdamW | AdamW | AdamW |
| | Batch Size | 256 | 128 | 64 |
| | Learning Rate | 0.001 | 0.001 | 0.001 |
| | Scheduler | cosine | cosine | cosine |
| | Weight Decay | 0.0002 | 0.0002 | 0.0002 |
| | Epoch | 100 | 180 | 180 |
| | $\|\epsilon\|_\infty$ | 0.3 | 0.031 | 0.031 |
| | $\alpha$ | 0.04 | 0.007 | 0.007 |
| | $\beta$ in TRADES | 1 | 6 | 6 |

# B  Related Works

## B.1  Robust Attacks and Defenses

The existence of adversarial examples [9, 10] has revealed the vulnerability of deep neural networks, where even imperceptible perturbations added to clean samples can deceive these networks. Such adversarial examples can be crafted by malicious adversaries in the real world [40, 41], thus necessitating models to possess sufficient adversarial robustness to effectively counter adversarial attacks.

To enhance the adversarial robustness of models against adversarial attacks, numerous adversarial defense methods have been proposed. Currently, the most widely used and effective defense method is AT [15, 16]. AT employs samples perturbed by adversarial attacks as training data, enabling the model to adapt to the distribution of adversarial examples and thus gain adversarial robustness [15, 16, 21]. Additionally, various other learning frameworks have been integrated into AT to improve its effectiveness, such as metric learning [42, 43], self-supervised learning [44, 45], and ensemble learning [46, 47]. However, AT still has two main limitations: 1) the inherent robustness-generalization trade-off in deep models [21], which leads to a decrease in the model's clean accuracy after AT; 2) RO [48, 49], where the model overfits to the adversarial examples in the training set.

## B.2  SSM Models

Due to SSMs' excellent long-sequence modeling capabilities and efficient linear computational complexity, these models have garnered widespread attention and achieved great success in various fields

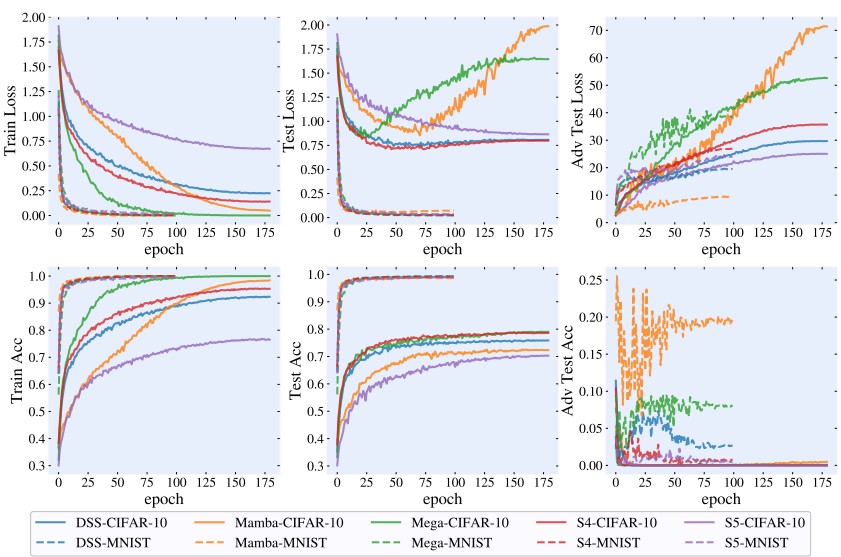

Figure 5: The ST training process, testing process, and the adversarial PGD-10 testing process on the testing datasets on CIFAR-10 and MNIST datasets.

Table 5: Comparisons among the test accuracy (%) of different SSM structures with different Training types on Tiny-Imagenet test set. 'Best' and 'Last' mean the test performance on the best and last checkpoint, respectively. 'Diff' denotes the accuracy gap between the 'Best' and 'Last'. The best checkpoint is selected based on the highest RA on the test set under PGD-10.

| Dataset | Structure | Training Type | Clean | | | PGD-10 | | | AA | | |
|---------|-----------|---------------|-------|-------|-------|--------|-------|-------|-------|-------|-------|
| | | | Best | Last | Diff | Best | Last | Diff | Best | Last | Diff |
| **Tiny-Imagenet** | S4 | ST | 21.00 | 20.88 | 0.12 | 0.00 | 0.00 | 0.00 | 0.00 | 0.00 | 0.00 |
| | | PGD-AT | 12.32 | 11.88 | 0.44 | 4.68 | 4.60 | 0.08 | 3.36 | 3.20 | 0.16 |
| | | TRADES | 17.92 | 17.44 | 0.48 | 3.48 | 3.24 | 0.24 | 3.00 | 2.88 | 0.12 |
| | DSS | ST | 23.20 | 23.00 | 0.20 | 0.40 | 0.00 | 0.40 | 0.00 | 0.00 | 0.00 |
| | | PGD-AT | 11.68 | 11.20 | 0.48 | 4.76 | 4.56 | 0.20 | 3.20 | 3.12 | 0.08 |
| | | TRADES | 17.24 | 16.96 | 0.28 | 3.28 | 3.08 | 0.20 | 2.64 | 2.40 | 0.24 |
| | S5 | ST | 18.64 | 18.60 | 0.04 | 0.00 | 0.00 | 0.04 | 0.00 | 0.00 | 0.00 |
| | | PGD-AT | 12.20 | 11.68 | 0.52 | 4.48 | 4.00 | 0.48 | 3.28 | 3.12 | 0.16 |
| | | TRADES | 14.72 | 13.96 | 0.76 | 2.56 | 2.20 | 0.36 | 2.12 | 1.92 | 0.20 |
| | Mega | ST | 36.84 | 36.80 | 0.04 | 1.60 | 0.00 | 1.60 | 0.00 | 0.00 | 0.00 |
| | | PGD-AT | 28.08 | 23.96 | 4.12 | 11.32 | 2.76 | 8.56 | 5.88 | 1.08 | 4.80 |
| | | TRADES | 32.12 | 30.00 | 2.12 | 10.44 | 8.48 | 1.96 | 5.24 | 4.88 | 0.36 |
| | Mamba | ST | 29.12 | 28.68 | 0.44 | 0.08 | 0.00 | 0.00 | 0.00 | 0.00 | 0.00 |
| | | PGD-AT | 27.52 | 27.36 | 0.16 | 6.48 | 4.04 | 2.44 | 4.08 | 1.92 | 2.16 |
| | | TRADES | 29.28 | 28.64 | 0.64 | 5.00 | 4.36 | 0.64 | 2.80 | 2.36 | 0.44 |

such as computer vision [50, 51] and natural language processing [20]. [1] was the first to integrate state-space models from modern control theory with deep learning, initiating extensive research on SSMs within the deep learning community. Early research on SSMs was limited by issues such as the computational complexity of training and the exponential decay of memory [52], which restricted their application. S4 [3] optimized the parameterization scheme of SSMs to reduce the computational requirements for training, while S5 [27] further extended S4's SISO system to a MIMO system. Mamba [20] introduced a selective scanning mechanism to more flexibly adapt to the input, significantly enhancing the performance of SSM models. Additionally, a series of works aimed to integrate SSM models with attention mechanisms [6, 53, 19] to mitigate the flexibility limitations imposed by the inherent fixed structure bias of SSMs.

### B.3 Adversrial Robustness of SSM Models

Despite the widespread attention on the superior performance of SSMs, the adversarial robustness of such models has not been thoroughly investigated. This is an urgent issue that needs to be explored before SSMs can be widely applied. To the best of our knowledge, [54] is the only work that has investigated the adversarial robustness of SSMs. This study shows that VMamba is more susceptible to adversarial attacks compared to convolutional networks on 2D image data, indicating that the robustness of SSMs requires more attention. However, this work only studied VMamba in the visual

Table 6: Comparisons among the test accuracy (%) of S4 and DSS with different AdS modules and different AT types on Tiny-Imagenet test set. 'Best' and 'Last' mean the test performance at the best and last epoch, respectively. 'Diff' denotes the accuracy gap between the 'Best' and 'Last', '–' indicates that the results of AdSS are not used. The best checkpoint is selected based on the highest RA on the test set under PGD-10.

| Dataset | Structure | Training Type | AdS | Clean | | | PGD-10 | | | AA | | |
|---|---|---|---|---|---|---|---|---|---|---|---|---|
| | | | | Best | Last | Diff | Best | Last | Diff | Best | Last | Diff |
| **Tiny-Imagenet** | S4 | PGD-AT | – | 12.32 | 11.88 | 0.44 | 4.68 | 4.60 | 0.08 | 3.36 | 3.20 | 0.16 |
| | | | ReLU | 12.64 | 12.36 | 0.28 | 5.02 | 4.92 | 0.12 | 3.56 | 3.44 | 0.12 |
| | | | Sigmoid | 12.47 | 12.12 | 0.35 | 4.89 | 4.82 | 0.07 | 3.43 | 3.34 | 0.09 |
| | | | Tanh | 12.58 | 12.33 | 0.25 | 4.95 | 4.85 | 0.10 | 3.52 | 3.43 | 0.09 |
| | | TRADES | – | 17.92 | 17.44 | 0.48 | 3.48 | 3.24 | 0.24 | 3.00 | 2.88 | 0.12 |
| | | | ReLU | 18.64 | 18.48 | 0.16 | 4.08 | 3.76 | 0.32 | 3.32 | 3.16 | 0.16 |
| | | | Sigmoid | 18.45 | 18.32 | 0.13 | 3.76 | 3.62 | 0.14 | 3.20 | 3.12 | 0.08 |
| | | | Tanh | 18.60 | 18.48 | 0.12 | 3.96 | 3.74 | 0.22 | 3.33 | 3.15 | 0.16 |
| | | FreeAT | – | 23.64 | 23.40 | 0.24 | 2.84 | 2.48 | 0.36 | 1.04 | 0.76 | 0.28 |
| | | | ReLU | 24.16 | 23.96 | 0.20 | 3.68 | 3.64 | 0.04 | 1.96 | 1.88 | 0.08 |
| | | | Sigmoid | 23.97 | 23.80 | 0.17 | 3.58 | 3.52 | 0.08 | 1.86 | 1.74 | 0.12 |
| | | | Tanh | 24.08 | 23.92 | 0.16 | 3.64 | 3.58 | 0.06 | 1.95 | 1.90 | 0.05 |
| | | YOPO | – | 18.68 | 16.08 | 2.60 | 5.12 | 4.88 | 0.24 | 3.04 | 2.92 | 0.10 |
| | | | ReLU | 19.60 | 18.44 | 1.16 | 5.72 | 5.52 | 0.20 | 3.64 | 3.48 | 0.16 |
| | | | Sigmoid | 19.41 | 18.57 | 0.84 | 5.62 | 5.47 | 0.15 | 3.55 | 3.39 | 0.16 |
| | | | Tanh | 19.62 | 18.49 | 1.13 | 5.69 | 5.57 | 0.12 | 3.68 | 3.51 | 0.17 |
| | DSS | PGD-AT | – | 11.68 | 11.20 | 0.48 | 4.76 | 4.56 | 0.20 | 3.20 | 3.12 | 0.08 |
| | | | ReLU | 12.04 | 11.88 | 0.16 | 5.08 | 4.92 | 0.16 | 3.84 | 3.68 | 0.16 |
| | | | Sigmoid | 11.87 | 11.65 | 0.12 | 4.96 | 4.81 | 0.15 | 3.75 | 3.60 | 0.15 |
| | | | Tanh | 12.13 | 11.95 | 0.18 | 5.11 | 4.96 | 0.15 | 3.92 | 3.73 | 0.19 |
| | | TRADES | – | 17.24 | 16.96 | 0.28 | 3.28 | 3.08 | 0.20 | 2.64 | 2.40 | 0.24 |
| | | | ReLU | 17.56 | 17.44 | 0.12 | 3.56 | 3.44 | 0.12 | 2.92 | 2.72 | 0.20 |
| | | | Sigmoid | 17.33 | 17.19 | 0.14 | 3.45 | 3.38 | 0.07 | 2.72 | 2.57 | 0.15 |
| | | | Tanh | 17.49 | 17.28 | 0.21 | 3.53 | 3.45 | 0.08 | 2.94 | 2.75 | 0.19 |
| | | FreeAT | – | 24.32 | 23.96 | 0.36 | 2.80 | 2.60 | 0.20 | 0.96 | 0.88 | 0.08 |
| | | | ReLU | 25.48 | 25.32 | 0.16 | 3.40 | 3.24 | 0.16 | 1.16 | 0.96 | 0.20 |
| | | | Sigmoid | 25.31 | 25.15 | 0.16 | 3.28 | 3.13 | 0.15 | 1.12 | 0.91 | 0.21 |
| | | | Tanh | 25.43 | 25.29 | 0.14 | 3.36 | 3.21 | 0.15 | 1.13 | 0.92 | 0.21 |
| | | YOPO | – | 18.76 | 17.44 | 1.32 | 5.28 | 5.08 | 0.20 | 3.20 | 3.04 | 0.16 |
| | | | ReLU | 19.76 | 18.92 | 0.84 | 5.92 | 5.80 | 0.12 | 3.40 | 3.20 | 0.20 |
| | | | Sigmoid | 19.48 | 18.78 | 0.70 | 5.72 | 5.66 | 0.06 | 3.32 | 3.15 | 0.17 |
| | | | Tanh | 19.70 | 18.87 | 0.83 | 5.85 | 5.77 | 0.08 | 3.39 | 3.18 | 0.21 |

domain and did not evaluate the sequence modeling robustness of SSMs. Additionally, it lacks a comprehensive evaluation of various SSM structures, and our understanding of the commonalities and characteristics of different SSM structures is still insufficient. More importantly, this study did not provide methods to improve the robustness of SSMs.

## C  Experimental Details

The specific structures of different SSM architectures on each datasets are shown in Table 3. And the training parameters and details used for each dataset are shown in Table 4. All experiments were implemented on multiple NVIDIA RTX A6000 GPUs.

## D  Additional Results

In this section, we report additional experimental results, including the ST training process, testing process, and the PGD-10 attack test results on the test datasets of CIFAR-10 and MNIST (Fig. 5), the CA and RA of five SSM structures on Tiny-ImageNet under different adversarial training frameworks (Tab. 5), and the related comparison of results with and without the AdSS mechanism (Tab. 6).

## E  Limitations

While this paper offers a componential analysis and exploration of SSM performance in AT, with the goal of informing the design of more robust SSM structures, there are several limitations to our study. Due to space constraints, our explorations have been limited to fundamental visual classification tasks. Future research will need to extend this analysis to a wider array of modalities and downstream tasks to fully understand and enhance the robustness of SSMs in diverse applications.

