# OpenReview forum: "Exploring Adversarial Robustness of Deep State Space Models"
_NeurIPS.cc/2024/Conference — NeurIPS 2024 poster_

### Official Review · Reviewer_a6wJ · 2024-06-25

**Soundness:** 3
**Presentation:** 2
**Contribution:** 2
**Rating:** 5
**Confidence:** 5

**Summary:**

The paper investigates and evaluates the effectiveness of traditional Adversarial Training (AT) methods on recently emerged State Space Models (SSMs). The paper finds that pure SSM is not a suitable choice for AT and attention-based SSM learns the adversarial feature more effectively. Based on such insights, theoretical proof is developed and the Adaptive Scaling is applied to SSM for better robust generalization. The experiments are mainly on MNIST and CIFAR-10.

**Strengths:**

1. The paper is well-written with a clear storyline and a suitable motivation.
2. The trustworthiness (e.g., robustness, explainability) research for SSM is an important topic and is yet to be fully explored.
3. The theoretical proof of the generalization bound is clear.

**Weaknesses:**

1. The evaluation dataset is not sufficiently scaled up to be representative. It would be more convincing if the paper developed similar findings on larger datasets such as CIFAR-100 and Tiny-ImageNet.

2. The involved adversarial training methods lack novelty. While PGD-AT and TRADES are classic AT methods, other representative variants can significantly improve AT efficiency and its robust performance. To name a few, Free-AT [1] substantially improves the training efficiency, and YOPO [2] boosts the robustness with fewer computations. Will SSMs be able to adopt these variants other than classic PGD-AT?

[1] Adversarial Training for Free. NeurIPS 2019.

[2] You Only Propagate Once: Accelerating Adversarial Training via Maximal Principle. NeurIPS 2019.

**Questions:**

Please address my concerns stated in the weakness section. Given the current version of the submission, I rate it as a borderline rejection considering the lack of sufficient empirical evaluation and the limited novelty of the adversarial training methods. However, I look forward to the authors' response, and I will consider revising the rating based on the soundness of the response.

**Limitations:**

The paper has stated the limitations in Appendix E.

---

> ### Author Response · Authors · 2024-08-06
> **Response to Reviewer a6wJ**
>
> **Response to W1**:
>
> Really greatful for the valuable comments from the reviewer. To ensure a more comprehensive and thorough assessment, we introduced a dataset with a larger size and more classes, Tiny Imagenet, for evaluation. The results are shown in the **Table 1 of the supplementary pdf**. The findings continue to support our discoveries on MNIST and CIFAR10, namely:
> 1) All SSM structures exhibit a significant drop in CA after AT, highlighting **the Robustness-Generalization Trade-Off present in SSM structures**.
> 2) Even the Data-Dependent Mamba model did not show a significant performance improvement compared to Data-Independent SSM structures like S4 and DSS after AT, indicating that **pure SSM structures struggle to benefit from adversarial training**.
> 3) **Although the incorporation of the attention mechanism has a positive effect on improving the model's CA and RA, our experiments on Tiny Imagenet also revealed potential RO issues it may bring**. For instance, the Mega model showed a significant drop in RA (8.56%) after PGD-AT, indicating noticeable RO in AT.
> 4) More importantly, **the addition of AdS still led to improvements in CA and RA for Data-Independent SSMs S4 and DSS on Tiny Imagenet, supporting the effectiveness of our AdS design**.
>
> Appreciate the reviewer once again for the valuable suggestions, and we believe this will further enrich the experimental evaluation of our paper and enhance its quality.
>
> **Response to W2**:
>
> We appreciate the constructive suggestions from the reviewer. **The purpose of this work is not to propose an AT strategy but to evaluate the AR of various SSM structures, analyze the component factors affecting SSM's benefit from AT, and guide the construction of corrective strategies**. Of course, incorporating more AT frameworks would help to refine the assessment of this work. Following the reviewer's suggestions, we introduced FreeAT[1] and YOPO[2], two more efficient AT frameworks, and conducted experiments on MNIST, CIFAR10, and Tiny ImageNet, with results shown in **Table 2 of the supplementary pdf**. The conclusions remain similar to our findings with PGD-AT and TRADES:
> 1) SSM structures under the FreeAT and YOPO AT frameworks also exhibit a decline in CA, **reflecting a clear trade-off between robustness and generalization**, especially on the CIFAR10 and Tiny Imagenet datasets, where this Trade-Off is more pronounced.
> 2) **Under the FreeAT and YOPO frameworks, pure SSM structures still struggle to benefit in terms of RA**. This result is consistent with our previous observations using PGD-AT and TRADES, indicating that these AT frameworks may not be optimized for SSM structures, or SSM structures themselves find it difficult to achieve effective robustness under these frameworks. Moreover, on the MNIST dataset, we noticed that all models had difficulty converging under the YOPO framework, which may suggest that certain characteristics of the YOPO framework are not fully compatible with the sequential image classification tasks of SSM structures. This finding indicates that the design of AT frameworks needs to consider the match between model structure and data characteristics more delicately.
> 3) Although the Mega model, which introduced an attention mechanism, showed higher CA and RA under the FreeAT and YOPO frameworks, its final RA decline compared to the best RA is also more significant, still **revealing the RO issues brought about by the introduction of the Attention mechanism**.
> 4) **When applying our AdS strategy under the FreeAT and YOPO frameworks, Data-Dependent SSMs S4 and DSS showed consistent improvements in both CA and RA**. This result further confirms the effectiveness and universality of the AdS strategy across different adversarial training frameworks.
>
> Thanks to the reviewer once again for the valuable suggestions, which are of great significance in enhancing the quality of our paper.
>
> [1] Adversarial Training for Free. NeurIPS 2019.
>
> [2] You Only Propagate Once: Accelerating Adversarial Training via Maximal Principle. NeurIPS 2019.

---

> > ### Comment · Reviewer_a6wJ · 2024-08-12
> >
> > Thank the authors for the discussion. The response is sufficient and valid. The arguments are reasonable with clear explanations. I revise the rating to 5 (Borderline Accept).

---

> > > ### Author Response · Authors · 2024-08-13
> > >
> > > Dear Reviewer a6wJ:
> > >
> > > We are honored by your thorough review. Your valuable suggestions have greatly enhanced our thinking and significantly improved the quality and depth of our research.
> > >
> > > Your insightful feedback has helped further elevate the quality of the paper. Additionally, your recommendation to include a discussion on more AT methods has been instrumental in refining the experimental evaluation of our work.
> > >
> > > We are particularly grateful for the professionalism and constructive feedback you demonstrated during the rebuttal phase. Your suggestions enabled us to improve the paper and ultimately gain your approval. This has been both encouraging and inspiring for us, and we believe these improvements will make our work more compelling.
> > >
> > > Sincerely,
> > >
> > > Authors of Paper #18034

---

### Official Review · Reviewer_PGjA · 2024-07-12

**Soundness:** 3
**Presentation:** 3
**Contribution:** 3
**Rating:** 6
**Confidence:** 3

**Summary:**

This paper investigate the adversarial robustness in deep state space models (SSMs). They provide both empirical and theoretical analysis of the SSMs' performance under the adversarial perturbation. They find that fixed-parameterized SSMs are limited in their adversarial training benefits due to output error bounds strictly tied to their parameters, whereas input-dependent SSMs risk experiencing error explosion.

**Strengths:**

1. The paper is well organized and easy to follow.
2. The experiments on serval image classification benchmarks are solid and comprehensive.
3. The theory analysis and visualization is helpful.

**Weaknesses:**

1. Tested dataset size is small.
2. Better to add a result of without AdS in Table 2 so we can better observe the improvement.
3. Lack of analysis why different activation function in AdS influence performance.

**Questions:**

1. Have you tested proposed AdS performance on larger dataset (e.g, ImageNet-1k, ImageNet-tiny)?
2. The proposed AdS is more like a naive try. Is that possible to propose a new SSM architecture based your observation?

**Limitations:**

See weaknesses and questions.

---

> ### Author Response · Authors · 2024-08-06
> **Response to Reviewer PGjA**
>
> **Response to W1**:
>
> Really appreciate the valuable feedback from the reviewer. To facilitate a more comprehensive assessment, we conducted evaluations on the tiny imagenet dataset, which has more classes and larger image sizes. The results, as shown in **Table 1 of the supplementary pdf**, are consistent with the conclusions observed on MNIST and CIFAR-10. Namely:
> 1) There is a clear Robustness-Generalization Trade-Off in SSMs. After AT, the CA of all SSM structures has significantly decreased. Particularly, the DSS showed a notably sharp decline in CA after PGD-AT, exceeding 10%, which significantly indicates the presence of a Robustness-Generalization Trade-Off.
> 2) Even the Data-Dependent SSM Mamba did not exhibit better robustness gains after AT compared to Data-Independent SSMs like S4 and DSS. This suggests that pure SSM structures struggle to benefit from AT.
> 3) While the introduction of the attention mechanism can improve the model's CA and RA, it also introduces the risk of RO issues. For instance, the Mega model showed an 8.56% decrease in final RA after PGD-AT compared to the best RA, which clearly indicates the existence of RO issues.
>
> We appreciate the constructive feedback from the reviewer once again.
>
> **Response to W2**:
>
> Very greatful for the constructive suggestions from the reviewer. In the revised version, we will report the results w/o AdS from Table 1 again in Table 2. Appreciate the reviewer's suggestions once again.
>
> **Response to W3**:
>
> Greatly appreciate the insightful comments from the reviewer. According to Theorem 1, when the eigenvalues of the state matrix $A$ are too large or too small, they can both lead to error accumulation during the state space transition: larger eigenvalues can increase the lower bound of the error, while smaller eigenvalues limit the model's expressive ability (for example, if the absolute value of the largest eigenvalue of $A$ is still close to 0, then the state transition will hardly preserve any sequence features). The sigmoid and tanh activations only have a shrinking regulatory effect, that is, when large eigenvalues of $A$ lead to excessive output errors, the use of AdS with Sigmoid or Tanh activation can reduce the error, but they do not have an amplifying function, so they are relatively limited in expressive power. On the other hand, AdS with ReLU activation has both shrinking and amplifying functions, thus it can achieve error reduction and alleviate the limitation of expressive power caused by small eigenvalues of $A$. We believe the above response addresses your concerns.
>
> **Response to Q1**:
>
> Greatful for the valuable question from the reviewer. Following the reviewer's suggestion, we have tested the performance of AdS on the larger dataset Tiny Imagenet, with the results also reported in **Table 1 of the supplementary pdf**. Specifically, even on the larger dataset, our AdS continues to provide universal improvements in CA and RA for Data-Dependent SSMs S4 and DSS.
>
> We appreciate the reviewer's suggestions once again, and we believe this will significantly enhancing the quality of our paper.
>
>
> **Response to Q2**:
>
> We appreciate the insightful questions from the reviewer. In this work, our aim is to explore the adversarial robustness of SSMs, to analyze the performance of SSMs in AT, to investigate the role of various SSM components in AT, and to provide further improvement measures based on the analysis.
> Specifically, by conducting a comprehensive comparison of the performance of various SSMs under AT, we:
> 1) First revealed the Robustness-Generalization Trade-Off in SSMs.
> 2) Further theoretical and experimental analysis explained why pure SSMs struggle to benefit from AT: the model's inherent state transition form leads to error accumulation, and the introduced attention mechanism plays a role in rescaling, thereby helping to mitigate errors introduced by perturbations.
> 3) Inspired by the above valuable conclusions and analysis, we considered introducing a low-complexity rescaling mechanism and proposed AdS. Further experiments on multiple datasets also demonstrated the universality and effectiveness of AdS.
> Therefore, the AdS we propose is not just a simple attempt; it is based on rigorous experiments and analysis.
> We believe that these findings and designs will provide insights for constructing more robust SSMs. However, the focus of this work is to explore the AR of SSMs and to analyze and provide improvement mechanisms, rather than proposing a new model structure. Of course, we think it could indeed help in designing new SSM architectures, but it requires 1) more refined design, including considering how to implicitly incorporate the idea of AdS into the SSM training framework (e.g. regularizing the state matrix with singular values) 2) extensive evaluation, including whether it can bring improvements to various vision and language tasks.
>
> We would express our gratitude to the reviewer once again for the valuable questions.

---

> ### Author Response · Authors · 2024-08-13
>
> Dear Reviewer PGjA,
>
> Thank you for taking the time out of your busy schedule to review our paper and provide valuable feedback. We greatly appreciate the issues you raised and have addressed them in detail. We have also made the following revisions to our manuscript based on your suggestions:
> Thank you for taking the time to review our paper and provide valuable feedback. We greatly appreciate the issues you raised and have addressed them in detail. Based on your suggestions, we have made the following revisions to our manuscript:
>
> **Evaluation on Larger Dataset**: Following your suggestion, we evaluated various SSM structures under different AT methods on the Tiny Imagenet dataset, which includes more categories and larger image sizes. The supplementary evaluation results are consistent with the conclusions observed on MNIST and CIFAR-10, further validating our findings.
>
> **Detailed Discussion of Theorem 1**: Based on Theorem 1, we have discussed the impact of the eigenvalues of the state matrix on error accumulation during state-space transformation. We also analyzed the influence of different activation functions on the model's expressive capacity, and have included this analysis in our paper.
>
> **Testing AdS Performance on Larger Dataset**: As per your suggestion, we tested the performance of AdS on the larger Tiny Imagenet dataset. The results demonstrate that AdS provides general improvements in CA and RA for data-dependent SSMs.
>
> **Table Refinement**: We have added the AT results without AdS to Table 2, as previously reported in Table 1.
>
> We believe these revisions contribute to improving the quality of our paper. If you have any further suggestions or questions, we would be happy to continue discussing and refining our work. Once again, thank you for the time and effort you have invested in this review.
>
> Sincerely,
>
> Authors of Paper #18034

---

> > ### Comment · Reviewer_PGjA · 2024-08-14
> >
> > Most of my concern has been addressed. I have increased my score to 6.

---

### Official Review · Reviewer_eBVC · 2024-07-21

**Soundness:** 2
**Presentation:** 3
**Contribution:** 2
**Rating:** 5
**Confidence:** 4

**Summary:**

This paper presents a comprehensive analysis of the adversarial robustness of Deep State Space Models (SSMs). The authors evaluate various SSM structures under different adversarial training (AT) frameworks, specifically examining how different components contribute to adversarial robustness. They observe that pure SSM structures struggle to benefit from AT, while the incorporation of attention mechanisms yields better trade-offs between robustness and generalization, albeit with the introduction of robust overfitting issues. The authors provide theoretical and empirical analyses to explain these phenomena and propose a simple Adaptive Scaling (AdS) mechanism to enhance SSM performance in AT.

**Strengths:**

- Originality: The paper addresses an important and under-explored area in the intersection of SSMs (that have become very prominent recently, both for text and vision) and adversarial robustness. It provides novel insights into how different SSM components behave under adversarial attacks and training, including important observations about robust overfitting.
- Quality: The work demonstrates high-quality research through its comprehensive empirical evaluations and some theoretical analysis. The authors conduct thorough comparisons across various SSM structures and AT frameworks.
- Clarity: The paper is well-structured and clearly written. The authors present their methodology, results, and analyses in a logical and easy-to-follow manner.
- Significance: This work makes significant contributions to understanding the adversarial robustness of SSMs, which is crucial given the increasing popularity of these models. The proposed AdS mechanism offers a practical solution to improve SSM robustness without incurring the drawbacks of attention mechanisms.
- Relation to previous works: The most related previous work is [10] which has conducted only preliminary robustness evaluations on visual SSMs (only VMamba architecture).

**Weaknesses:**

- I don’t quite understand the claim about robust overfitting: based on Table 1, using AutoAttack, the “Diff” is always very small, almost always <1%. Why is there robust overfitting then?
- Limited scope of experiments: While the paper provides comprehensive evaluations on MNIST and CIFAR-10 datasets, it lacks experiments on more complex datasets (e.g., at least Tiny ImageNet or some dataset with a higher image resolution) or real-world scenarios. This limitation somewhat restricts the generalizability of the findings.
- [Less important] Comparison with non-SSM models: The paper focuses exclusively on SSM variants without comparing their adversarial robustness to other popular model architectures like CNNs or Transformers. Such comparisons could provide valuable context for the robustness of SSMs relative to other widely-used architectures.

I would be willing to increase the score if the first 2 points are addressed.

**Questions:**

- It’s a bit surprising to see that PGD-AT doesn’t work on MNIST (according to AutoAttack), while TRADES does. Were the hyperparameters properly tuned?
- Have you considered exploring the interaction between SSM robustness and other aspects of model design, such as depth or width of the network?

**Limitations:**

Yes.

---

> ### Author Response · Authors · 2024-08-06
> **Response to Reviewer eBVC**
>
> **Response to W1**:
>
> Really appreciate the insightful comments from the reviewer. To ensure a fair evaluation, RO should be assessed under the same attack strategy, so we primarily judge RO based on RA on PGD-10. Our "Best" checkpoint is determined by the RA under PGD attacks. This aligns with the standard settings for exploring RO issues [1][2]. However, AA is a completely different attack strategy from PGD-10, making the "Diff" metric on AA inadequate for effectively measuring RO.
>
> We believe the above response will satisfactorily resolve the reviewer's questions.
>
> **Response to W2**:
>
> Greatly appreciate the insightful comments from the reviewer. To conduct a more comprehensive evaluation, we have performed ST and AT on various SSMs using the Tiny-Imagenet dataset, which offers more classes and larger image sizes. The results, as shown in **Table 1 of the supplementary pdf**, are consistent with the conclusions observed on MNIST and CIFAR10. Namely:
> 1)  All SSM structures experienced a decrease in CA on undisturbed data after AT. Notably, the DSS saw a CA drop of over 10% after PGD-AT, revealing a clear trade-off between robustness and generalization capabilities.
> 2) Even Data-Dependent SSMs like Mamba did not demonstrate a significant robustness improvement after AT compared to Data-Independent SSMs such as S4 and DSS. This indicates that pure SSM structures struggle to benefit from AT.
> 3) Incorporating attention mechanisms did indeed significantly enhance the model's CA and RA, but it also introduced RO issues. For example, the final RA of the Mega model decreased by 8.56% after PGD-AT compared to its optimal RA.
> 4) By integrating our AdS strategy, we were still able to universally improve the CA and RA for Data-Independent SSMs like S4 and DSS.
>
> We are grateful once again for the reviewer's insightful suggestions.
>
> **Response to W3**:
>
> We appreciate the insightful suggestions from the reviewer. The focus of this paper is to investigate the AR of SSMs based on their inherent structure and properties. To align with the research motivation of this paper, we have primarily conducted adversarial attack experiments on SSMs and their variants, and inspired by the experimental conclusions and theoretical analysis, we have designed mechanisms to enhance the robustness of SSMs. Therefore, we did not include baselines for CNNs and Transformers in the manuscript. Additionally, unlike traditional convolutional models such as ResNet that consider 2-D image classification, our setting is for sequential image classification, which precludes 2-D image classification CNNs from our comparison.
>
> Considering the aforementioned factors and the valuable input from the reviewer, we will include a set of experiments using Transformers for sequential image classification as a baseline in the revised version, to help readers better understand the performance of SSMs as sequence models. Thank the reviewer once again for the valuable suggestions.
>
> **Response to Q1**:
>
> Grateful for the insightful questions from the reviewer. We adopted the standard AT experimental setup, aligning with the configurations in [3][4]. In fact, we have experimented with adjusting the training hyperparameters (including learning rate, weight decay coefficient, etc.) but this did not alter the results. Since PGD-AT uses adversarial samples generated solely by PGD for AT, this could lead to overfitting to the distribution of adversarial samples produced by PGD, resulting in a significant decrease in RA against other attack strategies (such as AA), or even ineffectiveness. Similar experimental phenomena and conclusions are also found in [5]. We believe our response has adequately addressed your concerns.
>
> **Response to Q2**:
>
> We appreciate the valuable questions from the reviewer. Exploring the robustness of SSM across different scales of width and depth is an excellent question. As the focus of this work is to understand the behavior of SSM in AT from its inherent structure and to conduct an attribution analysis based on the impact of various SSM components, we have not considered the aspects of SSM width and depth. However, based on the theorems and experimental results of the paper, we also have some intuitive speculations: the error accumulation during the forward process of SSM, and due to the error accumulation effect of SSM, an increase in depth might actually constrain the robust generalization capability of SSM. We appreciate the insightful questions from the reviewer once again.
>
> [1] Overfitting in adversarially robust deep learning.
>
> [2] Exploring memorization in adversarial training.
>
> [3] Theoretically Principled Trade-off between Robustness and Accuracy.
>
> [4] Single-step Adversarial training with Dropout Scheduling.
>
> [5] Adversarial Training and Robustness for Multiple Perturbations.

---

> > ### Comment · Reviewer_eBVC · 2024-08-12
> > **Follow-up comments**
> >
> > Thanks for the new experiments and clarifications. The accuracy numbers on Tiny ImageNet seem quite low, though (e.g., best robust _accuracy_ with AutoAttack doesn't exceed 6%), but this is probably expected for non-convolutional architectures that are less sample-efficient compared to CNNs.
> >
> > As for robust overfitting, it should be assessed with the best possible attack (and AutoAttack is a decent proxy for that), not with the attack used for training. This affects many claims that you've made in your paper.
> >
> > I appreciate the new results, so I increase my score from 4 to 5. Overall, I feel like the paper provides a detailed study on Lp adversarial robustness in state-space models, but I'm not sure if there are some particularly interesting/unexpected messages in the paper, which is why I'm hesitant to increase my score above 5.

---

> > > ### Author Response · Authors · 2024-08-13
> > > **Follow-up Response to Reviewer eBVC**
> > >
> > > Thanks for the reviewers' feedback. We apologize for any doubts caused by our presentation and will address your questions point by point.
> > >
> > > **Q1: About the particularly interesting/unexpected messages in the paper**.
> > >
> > > **R1**: Apologize for any confusion caused by our presentation again. Our paper indeed reached some unexpected conclusions, and we will further elaborate on our conclusions:
> > >
> > > **Intuitive Understanding 1**: **Intuitively, Mamba's AT performance should be significantly better than that of S4 and DSS**. Mamba incorporates adaptive parameterization, allowing it to adjust SSM parameters adaptively according to the input, and thus should be more easily adaptable to perturbed inputs. Our conclusion contradicts such intuitive understanding.
> > >
> > > **Conclusion 1**: **Adaptive parameterization is detrimental to the AR of SSMs**.
> > >
> > > **Finding 1**: **The adaptive parameterization design of SSM in Mamba did not yield a performance gain over S4 and DSS in AT and even showed a negative gain** (See Table 1).
> > >
> > > **Analysis 1** (L240-L246): Our theoretical analysis (Lines 240-246), corroborated by validation experiments (See Fig.3), elucidates the reason behind the aforementioned finding: **adaptive parameterization design cannot ensure a bounded upper limit on the perturbation error of the SSM, while fixed parameterization can**. Therefore, the adaptive parameterization setting is detrimental to the AR of the SSM.
> > >
> > > **Intuitive Understanding 2**: **Intuitively, the AT performance after introducing Attention should be better than that of other models**. Since Mega with Attention has the best natural accuracy, and models with better natural accuracy generally perform better in AT [1][2], Mega's AT performance should be the best. However, our conclusion does not align with this intuition.
> > >
> > > **Conclusion 2**: **The introduction of Attention into SSM has brought about RO issues, which hinder the robust generalization capability of SSM**.
> > >
> > > **Finding 2**: **The incorporation of Attention has led to significant RO problems, greatly limiting the benefits that Attention could provide**. Particularly when using PGD-AT, the AT performance after introduced the Attention mechanism is lower than that of S4 and DSS (See Table 1).
> > >
> > > **Analysis 2**: The analysis from L249 to L291 indicates that the scaling effect of the Attention mechanism provides SSM with the ability to regulate perturbation errors, thereby improving AR. However, the integration of Attention introduces excessive model complexity, leading to RO issues that limit the gains brought by Attention.
> > >
> > > More importantly, **the findings and analysis above can also provide new insights for designing robust SSM architectures**, such as:
> > > 1) Introducing regularization into the design of adaptive SSM parameterization to prevent the unbounded growth of perturbation error brought by adaptive parameters.
> > > 2) Considering the introduction of adaptive scaling strategies with the lowest possible model complexity to replace the integration of Attention into SSM, thereby avoiding RO issues and further improving the AR of SSM.
> > >
> > > We will refine the presentation of our paper to more clearly convey the novelty and value of our conclusions.
> > >
> > > **Q2：RO should be assessed with the best possible attack**.
> > >
> > > **R2**: Thank you for the insightful comments from the reviewer. We fully agree with your suggestion to use AA for assessing AR. Accordingly, we conducted AR evaluations using AA attacks on various checkpoints of the models Mega and Mamba, which have RO issues, trained under the PGD-AT and TRADES frameworks, to measure the robust accuracy difference between the best and last checkpoints. The results are presented in the table below. The experimental outcomes indicate that both Mamba and Mega exhibit RO issues, with Mega showing a significant RO problem, especially when trained with TRADES on MNIST, where the robust accuracy difference between the best and last checkpoints reached $19.31$%. This is consistent with the conclusions provided in our paper. We will include the complete experimental information in the revised version.
> > >
> > > | Method | Model | MNIST (Best/Last/Diff) | CIFAR10 (Best/Last/Diff) | Tiny Imagenet (Best/Last/Diff) |
> > > |--------|-------|------------------------|--------------------------|--------------------------------|
> > > | PGD-AT | Mega  | 4.54/0.00/4.54          | 34.52/25.26/9.26          | 6.38/1.08/5.30                 |
> > > |        | Mamba | 2.13/0.00/2.13          | 37.29/32.28/5.01          | 4.15/1.92/2.23                 |
> > > | TRADES | Mega  | 30.21/10.90/19.31       | 41.48/36.97/4.51          | 6.44/4.88/1.56                 |
> > > |        | Mamba | 62.56/52.85/9.71        | 31.26/29.07/2.19          | 2.92/2.36/0.56                 |
> > >
> > > We believe that the above responses will fully resolve your concerns
> > >
> > > [1] Bag of Tricks For Adversarial Training.
> > >
> > > [2] Towards Efficient Adversarial Training on Vision  Transformers.

---

### Author Rebuttal · Authors · 2024-08-06

We are grateful for the comprehensive and professional feedback from the reviewers. It is both pleasing and encouraging to see that they have recognized our work as **novel** (R1), of **high quality** (R1), and of **significant importance** (R1, R3). They have also noted the **clear logic** in our writing (R1, R2, R3), a **well-defined motivation** (R3), **explicit theoretical contributions** (R2, R3), and **solid and comprehensive experimentation** (R2). We sincerely appreciate the reviewers' careful reading of our paper and their identification of the innovation, contributions, and strengths of our work.

We also thank the reviewers for their valuable and insightful comments and questions, which are crucial for further refining our work. We are confident that our responses will address the concerns raised by the reviewers.

Moving forward, we will first provide a unified response to the common questions from all reviewers, followed by a point-by-point reply to each individual's specific concerns.

**Q1: The evaluation dataset is limited and should be conducted on datasets with larger sizes/ scales to ensure more comprehensive evaluation.**

We greatly appreciate the valuable suggestions from the reviewers. To ensure a more thorough evaluation, we have conducted ST and AT on a larger dataset, Tiny Imagenet, which contains more classes. The results are presented in **Table 1 of the supplementary pdf**. Consistent conclusions with those obtained on MNIST and CIFAR10 can be drawn from these results, namely:
1) All SSM structures experienced a significant drop in CA after AT, especially the DSS, which saw a more than 10% decrease in natural accuracy after PGD-AT. This indicates **a clear trade-off between robustness and generalization in SSM structures**.
2) **Pure SSM structures hardly benefit from AT**. Even the Data-Dependent Mamba did not show a significant RA improvement compared to Data-Independent SSMs like S4 and DSS after AT.
3) **The incorporation of attention mechanisms indeed leads to better CA and RA, but introduces RO issues**, especially after PGD-AT, where Mega's final RA dropped by 8.56% compared to the best RA.
4) **After integrating our designed AdS, the CA and RA of Data-Independent SSMs, S4 and DSS, have been universally improved**.

Note that we have selected only the first 50 classes of the training and validation sets for experiments. The reason for this is the time cost required for experiments. We aim to better address the reviewers' questions within a limited time frame. For example, for models like Mega with a quadratic complexity attention mechanism, even using A800/A100 GPUs and a batch size of 128 on CIFAR10 for 180 epochs of TRADES AT, it takes more than three days. Tiny Imagenet's width and height are twice that of CIFAR10, which means the sequence length for sequential image classification will be four times that of CIFAR10, undoubtedly bringing a significantly higher time cost. Therefore, we chose the first 50 classes of Tiny Imagenet, ensuring that both the size and the number of classes of the dataset are significantly higher than those of CIFAR10 and MNIST for a more comprehensive evaluation. We believe our experiments and conclusions will effectively address the reviewers' concerns.

**Q2: More AT frameworks should be included for comparison to provide stronger support for the article's conclusions.**

We appreciate the constructive feedback from the reviewers. The purpose of this work is not to propose an AT strategy, but rather to conduct a comprehensive evaluation of the AR of SSMs. Additionally, by analyzing the structure of SSMs, we aim to identify the limitations that prevent them from benefiting from AT, thereby guiding the design of further corrective strategies. Considering the inclusion of newer AT frameworks to refine our assessment, we have introduced FreeAT[1] and YOPO[2], two more efficient AT baselines, and conducted experiments on MNIST, CIFAR10, and Tiny ImageNet. Hyperparameter settings were referenced from the original papers[1][2], with results shown in **Table 2 of the supplementary pdf**. We found consistent conclusions with those obtained using PGD-AT and TRADES:
1) Each SSM structure exhibited a reduction in normal accuracy, demonstrating **a clear trade-off between robustness and generalization**, especially on the CIFAR10 and Tiny ImageNet datasets.
2) **Pure SSM structures still struggle to benefit from these two AT frameworks**, and neither brought better RA than PGD-AT. Moreover, on MNIST, we found that all models had difficulty converging under the YOPO framework. This suggests that these two AT frameworks, effective for convolutional structures, may not be directly transferable to SSMs performing sequential image classification tasks. When using SSMs for sequential modeling, more targeted design is needed for the specific conditions of this scenario to analyze and design AT strategies.
3) Mega, which incorporates the Attention mechanism, showed a more pronounced decrease in final RA compared to other SSM structures, indicating a **more severe RO issue**.
More importantly, **the incorporation of our designed AdS into these two frameworks still universally improved the CA and RA for Data-Dependent SSMs S4 and DSS**. This further supports the effectiveness of the AdS design. We would like to express our gratitude once again to the reviewers for their constructive feedback.

[1] Adversarial Training for Free. NeurIPS 2019.

[2] You Only Propagate Once: Accelerating Adversarial Training via Maximal Principle. NeurIPS 2019.

---

### Decision · Program_Chairs · 2024-09-25

**Decision:**

Accept (poster)

**Comment:**

This paper presents a study of adversarial robustness of the state space models (SSMs), including both theoretical and empirical analysis. Through the analysis of different components of SSMs, the authors show that incorporation of the attention mechanism helps enhance their robustness due to its input-dependent nature, while there could be a risk of robust overfitting. Then, the authors proposed a simple mechanism to enhance the adversarial training performance while avoiding robust overfitting, and validated this on multiple small-scale benchmarks.

The paper received borderline ratings, with the reviewers leaning toward acceptance. The reviewers initially had concerns over the small-scale experiments, lack of experimental results with stronger attacks, underdeveloped method, and unclear claim over robust overfitting. However, these concerns were partly addressed during the rebuttal period as the authors provided clarification and additional results with more attacks and an additional dataset (TinyImageNet).

I believe that the paper proposes an interesting study that could be of interest to researchers working on SSMs, and the findings could lead to many follow-up studies in this direction. However, the problem addressed in this work seems niche, which may limit its potential impact.